# Measures Matter—Determining the True Nutri-Physiological Value of Feed Ingredients for Swine

**DOI:** 10.3390/ani11051259

**Published:** 2021-04-27

**Authors:** Gerald C. Shurson, Yuan-Tai Hung, Jae Cheol Jang, Pedro E. Urriola

**Affiliations:** Department of Animal Science, University of Minnesota, St. Paul, MN 55108, USA; hungx121@umn.edu (Y.-T.H.); jang0046@umn.edu (J.C.J.); urrio001@umn.edu (P.E.U.)

**Keywords:** amino acids, dietary fiber, digestion kinetics, functional ingredients, in vitro digestibility, lipids, minerals, prediction equations, swine, vitamins

## Abstract

**Simple Summary:**

Traditional proximate analysis measures (i.e., moisture, crude protein, crude fat, crude fiber, and ash) have little value for estimating the actual nutritional and economic value of feed ingredients fed to swine, yet they unfortunately continue to be commonly used in research studies, feed label regulations, and commodity-trading contract guarantees. Increasing energy and nutritional efficiency while simultaneously reducing negative environmental impacts of pork production requires the adoption of precision nutrition practices in global pork production systems. Precision swine nutrition can only be achieved by using more accurate and comprehensive methods and measures to determine the true nutri-physiological value of feed ingredients used in swine diets. There are several characteristics in some types of feed ingredients that are detrimental to swine health and performance, but they are seldom measured and deserve greater consideration. Likewise, there are also several value-added benefits provided by some functional feed ingredients that are not given enough consideration when formulating swine diets. The purpose of this review is to provide a holistic assessment of the benefits and limitations of existing chemical analysis methods and measures used to evaluate feed ingredients for swine and explore the benefits of using additional measurements to more accurately assess their true nutri-physiological value.

**Abstract:**

Many types of feed ingredients are used to provide energy and nutrients to meet the nutritional requirements of swine. However, the analytical methods and measures used to determine the true nutritional and physiological (“nutri-physiological”) value of feed ingredients affect the accuracy of predicting and achieving desired animal responses. Some chemical characteristics of feed ingredients are detrimental to pig health and performance, while functional components in other ingredients provide beneficial health effects beyond their nutritional value when included in complete swine diets. Traditional analytical procedures and measures are useful for determining energy and nutrient digestibility of feed ingredients, but do not adequately assess their true physiological or biological value. Prediction equations, along with ex vivo and in vitro methods, provide some benefits for assessing the nutri-physiological value of feed ingredients compared with in vivo determinations, but they also have some limitations. Determining the digestion kinetics of the different chemical components of feed ingredients, understanding how circadian rhythms affect feeding behavior and the gastrointestinal microbiome of pigs, and accounting for the functional properties of many feed ingredients in diet formulation are the emerging innovations that will facilitate improvements in precision swine nutrition and environmental sustainability in global pork-production systems.

## 1. Introduction

Multifaceted approach is emerging for selecting and sourcing feed ingredients in the global feed industry to achieve biosecure, econutritional, precision swine-feeding programs without the use of growth-promoting antibiotics. Historically, feed ingredients have been primarily selected based on availability and cost, as well as crude protein (CP), crude fat, and crude fiber content [1,2]. However, the urgent need to reduce the environmental impact of food animal production [3,4] has led to the creation of feed ingredient databases based on life cycle assessments (LCAs) of multiple environmental indicators [5,6]. These LCA environmental impact variables are being used as constraints when formulating “eco-diets” for swine to reduce environmental impacts of pork production [7,8,9]. In addition, international trade and travel have increased the risk of transboundary transmission of swine diseases [10], and feed-ingredient supply chains have been identified as a potential source of pathogen contamination and risk of swine disease transmission [11]. The discovery that swine viruses can survive in various feed ingredients [12], especially for African swine fever virus (ASFV) under the environmental conditions of transoceanic shipment [13], has led many swine-feed manufacturers to avoid sourcing feed ingredients from ASFV-infected countries [14,15]. As a result, biosecurity protocols are being developed and implemented to reduce the risk of ASFV contamination in feed supply chains [15,16]. A third dimension of feed-ingredient sourcing and selection involves the need to use alternative nonantibiotic nutritional strategies to support animal health. This consideration has evolved as a result of regulatory restrictions and prohibition of using growth-promoting antibiotics in animal feeds due to their contributions toward the development of antimicrobial resistance [17,18]. Therefore, exploiting the natural phytochemical properties of functional ingredients and nutrients as alternatives to antibiotic growth promoters [19,20] is emerging as a new approach for improving swine health.

All of these important criteria for selecting and sourcing feed ingredients must also be merged with precision swine nutrition practices, which are a key component of precision livestock farming. Precision livestock farming (i.e., precision animal-production systems) is a rapidly emerging, innovative, systems approach that is being implemented to improve productivity and sustainability of food animal production to meet the increasing need for safe and affordable food to feed a growing global population [21]. As precision animal production practices are adopted, farm profitability, efficiency, environmental sustainability, animal health, and welfare are also improved [22]. Precision animal nutrition and feeding is a core component of precision animal production systems, and involves providing customized diets that meet the changing nutrient requirements of animals over time while also accounting for differences in requirements among individual animals within contemporary groups [23]. However, achieving precision nutrition requires accurate determination of the true nutritional value and physiological responses of various feed ingredients used in diets formulated for precision feeding programs [24,25]. The ultimate goal of formulating precision nutrition diets for swine is to avoid oversupplying and undersupplying energy and nutrients relative to their daily requirements. Therefore, precision nutrition can only be accomplished by using nutritional measurements that accurately correspond to expected physiological responses, which we have described as the true “nutri-physiological” value of feed ingredients.

Although the determination of the true nutri-physiological value of feed ingredients and complete feeds has greatly improved over time, significant challenges still exist [25]. Initially, the Weende system of proximate analysis (i.e., moisture, CP, crude fiber, crude fat, nitrogen-free extract, and ash) was developed in an attempt to relate chemical composition of feed ingredients to their nutritional value at a time in history when very little was known about energy and nutrient digestibility of ingredients or nutritional requirements of animals [26,27]. Although the proximate analysis measures were useful for formulating animal diets at that time, they have since become woefully inadequate for optimizing caloric and nutritional efficiency in swine production systems today. However, because of their simplicity and widespread use around the world, proximate analysis components are still routinely measured and reported in nutrition research publications, and are used for establishing prices of commodity ingredients (e.g., corn, soybean meal, and distillers dried grains with solubles (DDGS)) in the global market. This is unfortunate because these measures do not accurately account for the true nutritional and economic value of ingredients. The actual nutritional and economic value of feed ingredients is only realized during least-cost diet formulation using metabolizable energy (ME) or net energy (NE), digestible amino acids (AAs), and digestible phosphorus content, not CP, crude fat, crude fiber, and ash content. As a result, using proximate analysis measures often limits the ability of buyers and sellers from capturing the true economic value of feed ingredients.

Fortunately, in vivo methodologies were developed to determine energy and nutrient digestibility of feed ingredients and complete feeds for swine, which was the beginning of estimating the actual nutritional contributions of feed ingredients from their chemical composition [28]. As new analytical methods and measures were subsequently developed and used in animal nutrition research, the contributions of nutrient fractions of feed ingredients toward body maintenance and productive purposes were better characterized and quantified [25]. More accurate quantification of energy and nutrient composition of feed ingredients, along with the evolution of expressing nutritional requirements of pigs from a total concentration basis to a digestible content basis, and subsequently to a bioavailable or net content basis, has greatly improved caloric and nutritional efficiency. As a result of these improvements, safety margins used in feed formulation can be reduced while feeding diets that more precisely meet the daily requirements of animals [25].

Despite these nutritional advances, some limitations of using conventional evaluation and analytical methods remain for assessing the utilization of dietary energy [29], AAs [30], and phosphorus [31]. Unfortunately, there are no standardized methodologies, or expressions of energy and nutrient digestibility, but recommended procedures for determining digestibility of several macronutrients for pigs have been reviewed and summarized [32]. Zhang and Adeola [33] also reviewed and compared techniques for evaluating digestibility of energy, AAs, phosphorus, and calcium in feed ingredients for pigs. Furthermore, nutrient content and digestibility estimates of various types of plant-based feed ingredients have been summarized [34]. Although all of these reviews are useful, they fail to explore additional chemical components that affect the true nutri-physiological value of feed ingredients beyond the use of traditional chemical measurements.

There are many challenges for determining the true nutri-physiological value of feed ingredients, including the development and standardization of accurate methods for estimating bioavailability of nutrients that are additive in mixtures of feed ingredients [30], and characterizing the metabolic fate of nutrients from various feed ingredients in animals [35]. However, now that liquid chromatography-mass spectroscopy (LC–MS) analytical capabilities are more widely available and used, there is a greater opportunity for more advances in precision animal nutrition through chemometric and metabolomic determinations [36,37,38]. In addition, new approaches for integrating metabolomics and other systems’ biology platforms with nutrition are emerging [39]. Dynamic, mechanistic, nutritional response models for predicting animal performance, whole-body energy and nutrient retention, nutritional efficiency of food products, and nutrient losses to the environment are also being developed [40]. The purpose of this review is to provide a comprehensive description of the challenges and opportunities for improving and refining our ability to connect chemical analysis with physiological and metabolic responses (i.e., nutri-physiological value) of various feed ingredients used in swine diets.

## 2. The “Disconnect” between Traditional Chemical Measurements and Physiological Responses

Feed ingredients are routinely evaluated both qualitatively and quantitatively to determine their perceived and real nutritional and economic value. Physical characteristics such as color, odor, and texture are subjective qualitative assessments that can be useful for initial screening of general acceptability of ingredient quality, but they are of limited value for assessing nutrient composition, digestibility, and presence of antinutritional factors. As a result, various chemical analyses have been developed and used to provide quantitative determinations of nutrient content, contaminants, and toxic compounds. However, results from chemical analyses vary among laboratories using the same procedure, as well as among different procedures used for measuring the same analyte [41]. In addition, chemical analyses provide no information regarding digestibility or expected physiological responses from various ingredients used in swine diets. Although an in vivo, whole-body, physiological evaluation is the most definitive method of determining the true feeding value of an ingredient, it is impractical for use in a quality control program for commercial feed manufacturing facilities because of the considerable cost and time involved in routinely conducting intensive controlled pig feeding experiments. Therefore, more thoughtful consideration is needed when selecting analytical methods and measures to ensure they are as closely associated with physiological responses as possible when assessing the true feeding value of various types of feed ingredients.

### 2.1. The Role of Water

Many animal nutritionists and feed manufacturers give little consideration to the moisture content of feed ingredients other than ensuring that it is low enough to avoid spoilage during long-term storage and to optimize pellet manufacturing conditions. Although measuring the moisture content of feed ingredients is generally considered to be simple and accurate, it is actually more difficult and complex than many nutritionists realize, and requires greater scrutiny because of its association with nutrient preservation and feeding value of ingredients. Moisture content can be determined using 35 different direct or indirect methods described by the Association of Official Analytical Collaboration (AOAC) International. Direct methods (e.g., air oven-drying, vacuum oven-drying, freeze-drying, Karl Fischer) are generally considered to be the most reliable measures but are more labor-intensive and time-consuming than indirect methods (e.g., NIR absorption, microwave adsorption, conductivity, refractometry). Furthermore, calculations of the percentage of moisture in a feed sample are often assumed to be interchangeable between reporting moisture on a wet or a dry basis. However, they are not interchangeable because when calculating moisture content on a wet basis, the amount of water is divided by the total weight of the sample (solids and moisture), while on a dry basis, the amount of water is divided by dry weight (only solids; [42]). Therefore, some of the variability in dry matter or moisture content of feed ingredients is due to the analytical method and type of calculation used. Although moisture content is an important nutritional measure, water activity (a_w_) may be even more important for understanding the nutri-physiological value of feed ingredients, and it is a commonly used measurement in human foods [43,44] and pet foods [45], but not in livestock and poultry diets.

Water activity is a measure of the energy state of water in food or feed and its potential to act as a solvent, participate in biochemical reactions, and support growth of microorganisms. The measurement of water activity was developed because moisture content of a food did not adequately represent the fluctuations in microbial growth that were commonly observed. Therefore, a_w_ is useful for predicting safety and stability of food and feed products because microbial growth is determined by water activity and not by water content, as commonly believed. Water activity is useful for predicting the types of microorganisms that may cause spoilage [43,45] and maintaining the chemical stability of feed ingredients. Water activity may also potentially be related to differences in pathogenic virus survival in various feed ingredient matrices [12]. Specifically, water activity in foods directly affects chemical and biochemical reaction rates such as spontaneous autocatalytic lipid oxidation reactions [46], nonenzymatic browning reactions [47,48], enzyme activity [49,50], and vitamin activity [51,52]. In addition, yeast and mold growth [53] and pellet-quality characteristics [54] are directly affected by water activity in foods.

It is unclear why the measurement of water activity is not conducted in feed ingredients used in animal diets to the same extent it is used in human foods, but it would be a useful indicator for determining if a feed ingredient is susceptible to microbial proliferation. Typically, a water-activity value below 0.60 is considered adequate for preventing bacteria and mold growth in foods [45], which is based on research results showing that a_w_ of 0.61 is the minimum value that allows fungal growth [53]. The relationship between moisture content and water activity of several grains at 25 °C varies, as shown in Table 1 [55]. Note that as moisture content of these grains increases, water activity also increases, but the incremental magnitude of this increase varies by grain type. This relationship should be considered when determining the maximum acceptable moisture content to prevent spoilage of grains during extended storage periods. Furthermore, particle size of common grains and oilseed meals appears to be an important factor affecting a_w_, where milling to a fine particle size appears to reduce water activity (Table 2 [54]).

### 2.2. Energy Systems

Energy is an essential component of animal diets because it is required for all types of body functions and processes. As a result, energy represents the greatest proportion of feed cost in animal production, which makes it critically important to use an energy system that precisely and accurately determines the proportion of gross energy (GE) of feeds that is utilized for maintenance and productive functions of animals [56]. The GE content of a feed ingredient is determined by measuring the amount of heat produced from combustion in an adiabatic bomb calorimeter and subsequently calculating the amount of energy released [56]. In general, the GE content of an ingredient is dependent on its carbohydrate, lipid, and protein content, but it is also influenced by carbohydrate composition (glucose = 3.75 kcal/g, starch = 4.16 kcal/g), as well as the fatty-acid and AA profiles (average of 5.64 kcal/g and 9.51 kcal/g, respectively; [57]). Apparent digestible energy (DE) is determined by subtracting the energy loss from the indigestible portion of the diet, intestinal cells, and endogenous secretions excreted in feces [56] from the GE consumed, and varies based on physiological stages of growth of pigs [58]. Metabolizable energy is defined as the proportion of DE content of a feed remaining after subtracting urinary and gaseous energy losses (mainly methane), but gaseous energy losses are usually not included in the calculation of ME content because they represent a relatively small percentage (0.1% to 3%) of DE [57].

Systems used for estimating energy requirements and utilization in feed ingredients for swine have evolved from DE to ME to net energy (NE) determinations. Net energy is the actual energy remaining for maintenance and productive purposes after accounting for additional energy losses from heat produced (heat increment) during ingestion, digestion, and physical activity beyond those attributed to ME [59]. The NE system is widely considered to be the most accurate approach for estimating the true energy value of a feed ingredient because it provides the best prediction of dietary energy contributions to pig growth performance [60]. The NE system avoids overestimating the energy value of protein and fiber, and underestimating the energy value of lipids and starch, while more accurately accounting for differences in metabolic utilization efficiencies between nutrients than DE and ME systems [61,62]. However, the use of the NE systems has not been widely adopted globally [56] because determining NE content of feed ingredients is more difficult than determining DE and ME content.

Even though NE systems are more accurate than ME and DE systems, an inherent assumption is that the energy used for body protein and lipid gain, as well as body maintenance functions, is not different between pigs with different rates and composition of body weight gain [63]. This assumption is not valid because the marginal efficiency of ME use for body lipid gain is greater than the marginal contributions of ME toward body protein gain and maintenance [63]. Mechanistic models have been developed to represent the contributions of dietary energy sources and use in pigs, but NE values of ingredients and nutrients are not constant and are affected by the rate and composition of pig growth [63]. Therefore, the authors of NRC (2012; [63]) developed the concept of “effective ME” and applied it to models used to estimate pig energy requirements. Using this approach, effective ME content of diets is calculated from NE content of the diet using fixed conversion efficiencies for starter pigs, growing-finishing pigs, or sows based on using corn and dehulled solvent-extracted soybean meal-based diets as reference diets and determining marginal efficiencies for various body functions. This approach allows the most accurate prediction of pig responses to energy intake when diet NE content is used as the input in the model to estimate effective ME content.

Several reviews have described the comparative advantages and disadvantages of using various methodologies for determining NE in feed ingredients [56,64,65,66,67]. Three NE systems have been developed, including the French system [62], the Dutch system [68], and the Danish system [69,70]. The French system determines NE estimates based on the energetic efficiency of starch, protein, and lipids from different nutritional sources for body protein and lipid deposition in growing pigs [71]. The Dutch system uses concentrations of digestible nutrients in feed ingredients to estimate the NE content, which is similar to the French system, but it also includes digestible and fermentable carbohydrate contributions [68]. In contrast, the Danish system is based on the concept of potential physiological energy released from adenosine triphosphate (ATP) bonds at the cellular level in pigs and uses a combination of in vitro digestibility and biochemical coefficients to estimate the potential ATP production from chemical components [69,70].

The ultimate goal of any energy system is to accurately rank feed ingredients based on their actual energy contribution for predicting pig performance. Because increasing amounts of high-fiber feed ingredients are being used in swine diets globally, use of the NE system is essential for avoiding overestimation of utilizable energy from these sources. In addition, use of the NE system minimizes the potential for overestimating energy utilization when formulating low-CP diets, which are becoming popular to minimize postweaning diarrhea and nitrogen excretion in manure for improved environmental sustainability. Because energy intake is the most limiting factor for protein deposition in young pigs, protein synthesis is an energetically expensive process, and energetic efficiencies to produce ATP or lipid synthesis are different [71], direct determination of NE content of more types and sources of ingredients is needed to develop and validate accurate equations for predicting pig responses from feeding diets with variable amounts of different types of energy-yielding nutrients.

#### 2.2.1. Carbohydrates

Carbohydrates are the primary chemical constituents of plant-based feed ingredients and serve as the primary energy source in swine diets. In addition, lactose and chitin are animal-derived carbohydrates that are present in a few feed ingredients fed to swine. Lactose is the main carbohydrate in milk for young mammals and serves as an energy source after it is hydrolyzed by lactase to glucose and galactose in the small intestine [72]. The addition of lactose to pig diets fed during the early postweaning stage of growth has been shown to improve nutrient digestibility and fermentability [73,74]. Lactose functions as a prebiotic for microbiota in the gastrointestinal tract [75] where lactic acid and volatile fatty acids are produced from partial fermentation [76]. In contrast, chitin is a structural polysaccharide present in insect exoskeletons and is classified as a carbohydrate, but it is indigestible for monogastric animals and is considered as a source of dietary fiber (DF; [77,78]).

Plant-based carbohydrates vary substantially in their monomeric composition, types of linkages, and extent of polymerization [79]. As a result, the relative contributions of different types of carbohydrates to NE vary based on their physical and chemical properties and physiological functions in swine diets. Monosaccharides, disaccharides, starch, and glycogen are all highly digestible forms of carbohydrates because they can be readily hydrolyzed in the small intestine, absorbed as monosaccharides, and used as sources of energy [80]. In contrast, carbohydrates that are not hydrolyzed by endogenous enzymes in the small intestine are referred to as DF [79]. The diverse nutritional and physiochemical properties of DF have resulted in the need to create an extensive classification system for carbohydrates. In fact, DF has been one of the most widely studied components of feed ingredients in recent years because of its: (1) abundance in many grain-based byproducts and coproducts [81,82,83]; (2) potential for hydrolysis from exogenous enzyme supplementation [84,85]; (3) fermentability and production of short-chain fatty acids that contribute toward energy requirements [86]; and (4) prebiotic effects on gut microbiome and improvements in gut health [87,88,89,90].

Because of the diversity of physicochemical characteristics of numerous DF components, it has been challenging to relate traditional chemical measurements for DF to specific physiological effects. The analytical methods most commonly used for determining DF content of feed ingredients and complete feeds include crude fiber, van Soest methods (neutral detergent fiber—NDF, acid detergent fiber—ADF, and acid detergent lignin—ADL), and total dietary fiber (TDF = soluble + insoluble dietary fiber) methods [83]. The crude-fiber method continues to be used in feed ingredient evaluation primarily because its measurement is highly reproducible within and among laboratories, which has made it acceptable for use by government regulatory agencies [83], for feed composition labels, and for commodity-trading purposes. However, crude fiber does not accurately characterize the DF portion of any complete feed or feed ingredient because of its incomplete recovery of cellulose, hemicelluloses, and lignin [83]. As a result, it is a poor measure for predicting nutritional and physiological responses from various types of DF for swine. The van Soest methods of NDF, ADF, and ADL [91] are better measures for connecting concentrations and characteristics of DF with physiological responses compared with crude fiber, but NDF and ADF underestimate TDF for high-starch ingredients because they do not account for soluble fiber components such as pectins, gums, and β-glucans [92]. The TDF method is more accurate than other methods of measuring DF because it quantifies the amount of soluble and insoluble components, including celluloses, hemicelluloses, some oligosaccharides, lignin, pectins, and gums [83]. Although partitioning DF into soluble and insoluble fractions in the TDF system aligns more closely with the physiological function of DF in monogastric animals, a single TDF value still does not adequately represent the complex physiological effects of different components of DF. As a result, DF sources are often characterized using nonstarch polysaccharide (NSP) composition based on individual insoluble (cellulose and some hemicelluloses) and soluble (pectins, gums, and β-glucans) components [93]. However, despite the greater association between TDF and NSP measures and physiological responses from DF, these measures have not been widely implemented in the feed industry because laboratory analysis is more complex, time-consuming, and expensive than the van Soest methods [83]. Therefore, instrumentation needs to be developed to overcome these challenges, or new analytical methods need to be created to better quantify the DF components that are associated with the physiological responses of DF for evaluating feed ingredients used in swine diets.

Dietary fiber comprises many carbohydrate components that have several physicochemical properties related to physiological responses. McRorie and Fahey [94] developed the most comprehensive classification system of connecting DF components to physiochemical properties, including: NSPs (Table 3), nondigestible oligosaccharides, other forms of plant-based and animal-derived fibers (Table 4), various forms of resistant starch, and chemically synthesized carbohydrate compounds (Table 5). These various DF types have been classified based on their physiological properties, including solubility (ability to dissolve in water), insolubility (inability to dissolve in water and remains as discrete particles), viscosity (ability to form high-molecular-weight gel when hydrated), and fermentability (extent of resisting digestion in the small intestine and to be degraded by microbiota in the cecum and colon to produce short-chain fatty acids and gas). Hydration properties of various fiber components were not specifically included as a factor in this classification system, even though properties such as swelling capacity, solubility, and water-holding and water-binding capacity may affect fermentability [95]. Several constituent groups comprise NSPs, including cellulose, hemicelluloses, β-glucans, pectins, gums and mucilages, and fructans; which provide distinctive functionality, including increased digesta viscosity and fermentability (Table 3). Cellulose and hemicelluloses are insoluble and poorly to moderately fermentable, while some soluble NSPs such as pectins, β-glucans, and gums increase digesta viscosity. Classification of feed ingredients based on these physicochemical characteristics of DF is becoming essential for predicting the energy contribution, effectiveness of exogenous enzymes, and prebiotic effects of diets containing various high-fiber ingredients [83]. In fact, recent studies have suggested that formulating swine diets containing high-fiber ingredients should be based on viscosity constraints [96], and prediction equations can be used to estimate energy contributions of short-chain fatty acids produced from DF fermentation [86].

There is increasing interest and evidence indicating that the time of day when DF fermentation occurs is an important determinant of its utilization. Circadian rhythms affect eating behaviors, gut microbiome, nutrient utilization, and health of animals [97]. Most physiological processes are controlled by daily circadian rhythm cycles, in which feed consumption activity serves as a major signal for synchronizing biochemical activities, DF fermentation rates, and modifying nutrient delivery rates that influence growth rate, energy partitioning, reproduction, and animal well-being. Therefore, we must develop measurements and approaches to improve our ability for matching the nutrient requirements of the gut microbiome at various stages of the daily circadian rhythm cycle to ensure proper timing of nutrient delivery to optimize their utilization [98,99,100]. This can be achieved by knowing the rate and extent of DF fermentation at specific locations along the gastrointestinal tract that relate to the microbiome and other physiological effects.

The physicochemical properties of DF can have detrimental or beneficial effects in swine diets. Numerous studies have shown that high-fiber diets generally decrease nutrient digestibility and increase endogenous nutrient loss, leading to a detrimental effect on energy metabolism and growth performance of pigs [83]. Viscosity of DF, in addition to its concentration in the diet, has been shown to have a substantial effect on decreasing nutrient digestibility of pigs [96]. Soluble NSP is associated with increased digesta viscosity and undesirable fermentation that creates an environmental niche for pathogenic bacteria growth, such as *Escherichia coli* [101,102,103]. As a result, gut health of pigs is eventually impaired. In contrast, feeding diets containing slowly or poorly fermentable DF (i.e., insoluble fiber) has been shown to cause laxative effects and a shorter retention time in the gastrointestinal tract, which improves fecal consistency and gut health due to less time for the proliferation of pathogenic bacteria and accumulation of undigested material for fermentation in the lower gastrointestinal tract [103,104,105]. Furthermore, poorly fermentable DF from wheat straw and DDGS has been shown to shift intestinal epithelial-cell differentiation toward secretory cells, where goblet cells can protect the gut epithelium, but may compromise nutrient absorption [106]. Rapidly fermented DF types serve as nutrient and energy sources for the gut microbiota by increasing production of short-chain fatty acids while decreasing ammonia concentration to promote animal health [82,90]. However, excess fermentable DF can cause osmotic imbalance and other intestinal disorders [103,107]. Additional studies are needed to provide greater insights into the complexity and interactive physiochemical properties of various forms of DF and their nutri-physiological contributions to swine diets.

Feeding high-fiber diets to sows provides several physiological and reproductive benefits [108]. Limit-feeding high-fiber diets can improve postprandial satiety, but the fermentation characteristics of the type of DF provided determines the relative effectiveness of this response [109]. In addition, feeding diets containing some types of DF prior to mating have been shown to improve embryo survival after mating in gilts [110,111]. Feeding diets containing high concentrations of fermentable NSPs to sows from weaning to estrus and during subsequent gestation periods for three reproductive cycles has been shown to increase the total piglets born and the born-alive litter size [112]. Studies have also shown that feeding certain types of high-fiber diets during specific phases of the reproductive cycle improves litter birth weights and weaning weights [113,114]. Collectively, although results from these studies indicate that feeding certain types and amounts of DF from common ingredients during specific phases of gilt and sow reproductive cycles can result in improved reproductive performance responses, results from other studies have not shown these benefits [108]. Therefore, future advances in capturing the reproductive benefits from feeding high-fiber diets to sows will depend on our ability to connect physiochemical characteristics of different types and concentrations of DF with these types of physiological and reproductive responses.

#### 2.2.2. Lipids

Lipids contain 2.25 times more energy than carbohydrates, but crude fat (ether extract; EE) content is poorly associated with ME and NE content of grains [115,116], as well as grain and oilseed byproducts [117,118]. There are several reasons for this. First, crude fat is determined using an EE procedure, which does not completely extract all lipids, especially if they are present as salts of divalent cations or are linked to carbohydrates or proteins [119]. Therefore, EE underestimates the true lipid content of feed ingredients. However, the addition of an acid-hydrolysis step to the extraction process results in the release of more of the lipids bound to carbohydrates and proteins, as well as sterols and phospholipids, and provides a more accurate method of quantifying true lipid content of feed ingredients [120]. As a result, the lipid content in various feed ingredients is generally greater using acid-hydrolyzed ether extract (AHEE) than EE [63,119,121], but this is not always true [122]. Second, it is important to recognize that true ileal and total tract digestibility of AHEE in extracted oil is greater than in byproducts containing intact sources of corn oil or soybean oil [123]. Endogenous losses of lipids are affected by differences in the fatty acid composition of dietary lipids, as well as the concentration and source of NDF in diets fed to growing pigs [124,125]. Third, the EE measurement provides no information about the fatty acid profile of lipids. The age of the pig postweaning [126,127,128], fatty acid chain length [129,130], degree of unsaturation [121], free fatty acid (FFA) content [129,131,132,133], fatty acid position on the glycerol molecule of triglycerides [134,135], and relative proportions of various fatty acids in a lipid source affect digestibility and ME and NE value. However, studies that have evaluated and compared differences in digestibility among lipid sources based on age of pig are limited [58,136]. Finally, unsaturated fatty acids (UFA) are susceptible to oxidation, which may reduce energy value of some lipids [137], but not others [138]. Thermal processing of lipids decreases digestibility by oxidizing the double bonds in UFAs, which increases oxidation products that can either disrupt the activity of pancreatic lipase or be polarized and poorly absorbed [139].

One of the greatest challenges of capturing full nutritional and economic value of any feed ingredient is to use accurate energy values for specific sources used in swine diet formulations. Feed fats and oils contribute a substantial proportion of energy to swine diets, but like other feed ingredients, the chemical composition and DE, ME, and NE content is highly variable within and among sources [140]. For example, soybean oil, which is one of the most abundant and commonly used lipid sources in swine diets around the world, has a DE content ranging from 7977 to 9979 kcal/kg, and a ME content ranging from 7906 to 8868 kcal/kg (Table 6). Age of pig and diet inclusion rates of supplemental lipids are two of the major factors contributing to variability in energy content. Although there are fewer published estimates of NE content for soybean oil and other fats and oils for swine compared with DE and ME content, reported NE values also highly variable—where the NE content reported for soybean oil ranges from 4561 to 8132 kcal/kg among published studies (Table 7). This creates a dilemma for nutritionists when deciding on which DE, ME, or NE value to assign to the specific lipid source when formulating swine diets. This challenge may be overcome if accurate prediction equations can be developed and used to relate chemical composition of lipids to DE, ME, and NE content.

Fortunately, there continues to be considerable interest in developing and using robust equations to accurately estimate DE, ME, and NE content of lipids based on chemical composition (Table 8). However, although several equations have been developed for swine, their accuracy in predicting in vivo determined DE content is generally poor [141,144,150]. Kerr et al. [150] determined the DE content of five sources of distillers corn oil (DCO) containing 0.04% to 93.8% FFA, and compared these values with predicted DE content using equations by Wiseman et al. [151], which included FFA content and UFA to saturated fatty acid (SFA) ratio as the main variables (Table 9). The predicted DE content of these lipids averaged about 300 kcal/kg more than the in vivo determined DE content for all DCO sources except for the source containing 93.8% FFA, which was predicted to have 1100 kcal/kg more than was actually determined in vivo (Table 9). Kellner and Patience [141] determined the DE content of 14 different lipids, including two sources each of choice white grease, corn oil, and soybean oil fed to 13 kg and 50 kg pigs, and compared these in vivo values with predicted values using equations from Powles et al. [129] and equations derived from chemical composition of lipids evaluated in their study (Table 10). Their results showed that the differences in chemical composition among lipid sources resulted in different energy values depending on the body weight of pigs. Furthermore, the Powles et al. [129] equation underestimated the negative effect of FFA content and was not accurate for estimating DE content of more saturated lipid sources that had a high proportion of fatty acids with chain lengths less than 16 carbons. In a subsequent study, Kerr et al. [144] determined the DE content of 10 different fats and oils for 19 kg pigs (Table 11), and reported values ranging from 8071 kcal/kg (tallow) to 9979 kcal/kg (soybean oil). However, when using the Powles et al. [129] prediction equation to estimate DE content based on chemical composition of these lipids, and comparing predicted values with in vivo determined values, DE content was underestimated by 243 kcal/kg (tallow) to 2276 kcal/kg (coconut oil), except for fish oil, which was overestimated by 289 kcal/kg (Table 11). When Kerr et al. [144] used the Kellner and Patience [141] DE prediction equations in a similar comparison, predicted DE content was also underestimated (440 to 967 kcal/kg), but generally to a lesser extent than the corresponding DE estimates derived from the Powles et al. [129] equations, while the DE content of tallow was overestimated by 393 kcal/kg.

There may be several reasons for the lack of accurate predictions of DE content observed in these studies. First, the equations are likely to be too simplistic and do not include other important predictive variables that may improve their accuracy and potential use. Second, if prediction equations are used to estimate energy content of fats and oils, they should be derived from experiments that have directly determined DE, ME, or NE content of those specific types of lipids, and not from studies that evaluated other types of ingredients or different lipid sources. Third, energy prediction equations derived from various lipids with diverse fatty acid profiles are assumed to provide reasonably accurate predictions when compared with energy values directly determined from in vivo experiments. However, results from Kellner and Patience [141] and Kerr et al. [144] have shown that this often does not occur. There may be at least two explanations for this.

Although prediction equations may reasonably estimate DE, ME, or NE content of lipid sources in an experiment from which they were derived [141], they may not result in accurate predictions when applied to lipid sources and composition not included in the original data set [144]. In addition, most energy prediction equations have been derived without considering the extent of oxidation of the lipids evaluated. Lipid oxidation has been shown to reduce energy content of lipids, but not always [138,152]. In fact, Rosero et al. [133] concluded that prediction equations may be improved by including measures of lipid oxidation, even though FFA concentration and extent of saturation of lipid sources explained a large proportion of the variation in DE content of lipids.

Feed fats and oils, as well as the lipid fraction in other feed ingredients such as corn DDGS [153], are frequently exposed to pro-oxidants (heat, moisture, oxygen, light, and transition metals) during processing and storage that cause lipid oxidation [154]. Lipid sources containing high proportions of UFAs are more susceptible to oxidation than SFA sources, and several oxidation products, including peroxides, aldehydes, ketones, acids, esters, hydrocarbons, epoxides, polymers, lactones, furans, and aromatic compounds, are produced during various stages of the oxidation process [155,156,157,158]. Many peroxidation indicator assays can be used to assess the extent of oxidation in lipid sources, but none of these measures provide a comprehensive assessment of all types and concentrations of oxidation products and their effects on animal health and performance [154]. Hung et al. [159] conducted a meta-analysis of published data from poultry and swine studies and reported that feeding oxidized lipids resulted in reduced average daily gain (ADG; 5%), average daily feed intake (ADFI; 3%), gain:feed (2%), and serum or plasma vitamin E content (52%), while increasing serum thiobarbituric acid reactive substances (TBARS) concentration by 120% compared with animals fed unoxidized lipids. These results clearly indicate that feeding oxidized lipids contributes to increased oxidative stress, but peroxide value (a measure of peroxidation) was not correlated, and dietary TBARS concentration was only moderately negatively correlated with ADG (r = −0.58) in pigs. Because feeding diets containing oxidized lipids contribute toward increasing oxidative stress [160,161], mortality [162,163,164], and impairing immune function [165,166], new analytical approaches are needed to quantify and connect the many complex chemical components of lipids and develop more robust equations to more accurately estimate the DE, ME, and NE content of fats and oils used in swine diets.

### 2.3. Protein and Amino Acids

In the Weende analysis system, CP is defined as the nitrogen-containing fraction of a feed ingredient [26,27]. Crude protein content is estimated by multiplying the nitrogen content by a constant factor of 6.25, which is the inverse of 16% of the assumed weighted average nitrogen content of proteins. However, this approach is not valid because AA profiles and amounts of nonprotein nitrogen vary among ingredients. Nonprotein nitrogen compounds, such as nucleic acids and nucleotides, some vitamins (e.g., thiamin), amines, amides, and urea can contribute a significant proportion of the nitrogen represented in CP that is not associated with proteins and amino acids. Furthermore, the CP measure provides no information regarding the concentrations, proportions, digestibility, and bioavailability of AAs in feed ingredients, which is essential information for accurately formulating swine diets. Despite these many limitations, CP is still widely used in feed ingredient marketing and feed label regulations as an indicator of nutritional and economic value.

Because protein is the second most expensive component of swine diets, and represents a significant portion of the total diet, accurate estimation of AA content in feed ingredients is needed to make real-time adjustments in databases used for precision swine-diet formulation [167]. In general, poor correlations between CP and lysine content have been reported for corn, soybean meal with and without hulls [41], corn DDGS [168], and wheat middlings [169], but correlations between CP and other indispensable AAs are often much greater. Messad et al. [170] recognized the limitations of using simple regression analysis for developing equations to predict AA content from CP content, and used a meta-analysis approach. In their analysis, data on feed-ingredient composition from 34 studies conducted between 1977 to 2012 were used to develop AA prediction equations for faba beans, peas, lupins, soybeans, wheat, barley, corn, sorghum, decorticated oats, corn DDGS, soybean meal, and rapeseed meal. Although several of the models developed in this study were shown to be relatively accurate for estimating the indispensable AA content from CP for many common feed ingredients used in swine diets, using CP to estimate AA content in feed ingredients is inadequate for precision swine feeding programs [170]. In addition, estimating concentrations of dispensable (i.e., nonessential) AAs from CP content of feed ingredients has generally been considered to be of minimal value in most previous studies because of the assumption that they are synthesized in sufficient quantities to meet the animal needs for maximal growth and optimal health. However, there is no longer strong evidence that supports this assumption because of several important functional roles that many of these dispensable AAs provide in swine diets [171].

More importantly, nutritionists need methods to dynamically estimate digestible and bioavailable AA content of feed ingredients used in precision diet formulation. Several reviews have been published describing the advantages and disadvantages of using various in vivo experimental techniques and procedures for determining AA digestibility in feed ingredients for swine [32,33,172]. Standardized ileal digestibility (SID) is the most accurate measure of expressing AA digestibility in feed ingredients because it corrects for basal endogenous losses of AAs during digestion [173], and SID values are additive in mixed diets, which allows them to be used in practical swine diet formulations [174]. The use of SID values for AAs in commercial swine diet formulations has greatly improved the accuracy of predicting animal performance responses [175]. However, determining SID values requires that in vivo experiments be conducted, which involve usingspecialized surgeries to insert ileal cannulas that are expensive, time-consuming, and impractical for commercial feed manufacturers to conduct. Furthermore, the digestibility values obtained in these in vivo digestibility experiments are accurate only for the specific feed ingredient sources evaluated. As a result, various in vitro procedures have been developed and evaluated for accuracy, especially for heat-processed feed ingredients such as soybean meal and corn DDGS.

When feed ingredients are subjected to heat treatment during processing, the epsilon amino group of free and protein-bound lysine may react with reducing sugars to create several types of Maillard products [176,177]. In the early stages of thermal exposure, structurally altered lysine derivatives called Amadori compounds are produced, which interfere with AA analysis and result in inaccurate determination of lysine content in feed ingredients. Lysine that is bound to these derivatives is often described as “blocked lysine” because it is biologically indigestible and unavailable for use by monogastric animals [178]. However, during the analysis procedure, which involves acid hydrolysis, up to 50% of blocked lysine is released and detected as lysine [179], while the remainder is released as furosine and pyridosine. During the late stages of thermal processing of a feed ingredient, melanoidins are produced but are not measured in AA analysis procedures, and results in a lower calculated lysine-to-CP ratio. As a result, furosine [180], reactive lysine [104,181,182,183], acid detergent insoluble nitrogen, and lysine-to-CP ratio [184] have been suggested as indicator measures of lysine digestibility in heat-processed feed ingredients (i.e., soybean meal and DDGS). Although color of DDGS was initially thought to be a reasonable indicator of lysine digestibility in DDGS [185,186], results from a more robust subsequent study showed a poor association between DDGS color and SID of lysine and other AAs [187]. In contrast, optical density and front-face fluorescence analytical techniques appear to be promising methods for rapidly estimating SID of AAs in DDGS, but the accuracy of these procedures has not been validated [187]. In soybean meal, urease activity [188], protein dispersibility index [189], and KOH solubility [190,191] have been used extensively to determine the quality and digestibility of CP, but all of these methods have significant limitations and inaccuracies [192,193,194]. However, unlike most of the in vitro methods, the use of advanced near-infrared spectroscopy (NIRS) appears to be a promising method for rapidly, inexpensively, and accurately determining reactive lysine in oilseed meals and AA digestibility in feed ingredients used in swine diets [195].

Development and use of prediction equations to estimate SID of AAs in feed ingredients based on chemical composition has also been considered as an alternative approach to overcome the challenges of using the various in vitro methods. Zeng et al. [196] conducted a meta-analysis of published chemical composition and SID AA data for corn DDGS, and developed simple prediction equations that consisted of the total AA concentration and either NDF or ADF content to accurately predict the SID AA content of corn DDGS sources for growing pigs. Using a meta-analysis approach, the accuracy and precision of these prediction equations were greatly improved compared to those from previously published studies [184,185,186]. Similarly, Messad et al. [170] used a meta-analysis approach to predict apparent ileal digestibility (AID) of AAs from total AA content, and also developed equations to predict SID of AAs from AID values for 12 common feed ingredients. Results from this study showed a relatively good positive linear association (R^2^ = 0.72 to 0.88) between total concentrations of all indispensable AAs and AID of AAs, but correlations differed between ingredients due to the negative effect of NDF content on AID in some ingredients. Although the prediction accuracy of all new SID equations developed was improved compared with original versions, they require validation before being used in practical swine diet formulations [170]. Other approaches have also been evaluated that include usingprediction equations requiring in vivo AA digestibility data derived from a rat model, along with data obtained from two- or three-step in vitro methods, to provide reasonable precision in estimating SID of AAs of feed ingredients for pigs [197].

Perhaps the most promising approach for predicting biological responses from chemical composition of feed ingredients involves using LC-MS, advanced proteomic, and bioinformatic approaches [197]. The nutritional, chemical composition, functional value, and bioactivity of proteins in six different feed ingredients (casein, partially delactosed whey powder, spray-dried porcine plasma, soybean meal, wheat gluten meal, and yellow meal worm) have been determined and compared using an untargeted LC-MS approach [198]. The use of untargeted chemometrics allows for determining both qualitative and quantitative information on the protein molecules present in feed ingredients and their potential functional properties. The combined use of proteomic and bioinformatic approaches may be useful for developing methods to improve protein quality assessment of feed ingredients, monitor and determine the effects of processing on protein digestibility and bioavailability, and evaluate digestion kinetics of various protein ingredients [198]. Furthermore, LC-MS analytical platforms are necessary to detect protein oxidation biomarkers when evaluating protein quality of feed ingredients [199].

Protein oxidation of feed ingredients is an emerging area of concern because of its potential contributions toward exacerbating oxidative stress in animals [199]. Protein oxidation occurs when a protein is covalently modified directly by reactive oxygen species or indirectly by reactions of secondary metabolites of oxidative alterations [200]. Unlike lipid oxidation, our current understanding of the extent of protein oxidation in feed ingredients is limited, and its potential dietary impacts on swine health and performance are poorly understood. Protein oxidation can occur by heating and grinding of feed ingredients [201,202]. Several in vitro and in vivo studies have shown that heat-induced protein oxidation directly alters the structure, affects the functional properties, and reduces antioxidant properties in soy protein isolate [203,204,205]. Feeding soybean meal containing heat-induced oxidized protein to broilers resulted in reduced growth performance and antioxidant status [206]. The impact of storage time and temperature on protein oxidation of several rendered animal byproducts has been evaluated, and an increase in carbonyl content was observed in chicken blood meal and beef meat and bone meal stored at 45 °C for seven days, while fish meal and chicken blood meal had increased carbonyl concentration when stored at 20 °C for six months [207]. Feeding diets containing oxidized spray-dried porcine plasma to weaned pigs for 19 days reduced protein digestibility and increased crypt depth in the small intestine but had no effect on measures of oxidative stress [208]. Similarly, feeding diets containing oxidized proteins and lipids from chicken byproduct meal to weaned pigs resulted in reduced energy and nutrient digestibility and growth performance, but did not contribute to oxidative stress [208]. Future studies are needed to determine the extent of protein oxidation in various thermally processed feed ingredients, and the significance of dietary oxidized proteins on oxidative stress, health, and performance of swine.

### 2.4. Minerals

Phosphorus is the third most expensive component of swine diets and requires special consideration for optimizing its utilization in swine diets due to its potential adverse effects on the environment. Calcium and phosphorus are the most abundant minerals in animals, and about 90% of Ca and 80% of P in the body is concentrated in the skeleton [209]. Feed ingredient sources of Ca and P are classified as organic (plant and animal) and inorganic in origin, with animal-derived and inorganic sources containing the greatest concentrations and digestibility of these minerals. In contrast, plant-based ingredients generally have low concentrations of Ca and low to moderate concentration of P, which is poorly digestible because the majority is in the chemical form of phytate and is indigestible for swine [63].

Historically, P bioavailability estimates were used for assessing the proportion of total P that was digested and absorbed by the animal [210,211]. However, because the slope-ratio procedure used to determine these estimates involves using an inorganic P source as a reference, the resulting values obtained were actually “relative” bioavailability estimates of P [210]. Therefore, because the P bioavailability is not 100% in the inorganic P reference source, relative bioavailability does not accurately represent true bioavailability [212]. If this is not considered when formulating diets on an “available” phosphorus basis, the actual amount of dietary P that will be utilized by the pig will be overestimated and will likely result in a P deficiency. In fact, the actual utilization of P in corn has been underestimated by 35% in growing pigs due to the use of published values for apparent digestibility and bioavailability of P [213].

To overcome this problem, methodologies to determine apparent (ATTD), standardized (STTD), and true total tract digestibility (TTTD) of Ca and P have been used to more accurately express their utilization in various feed ingredients for swine [32,33]. Unlike ATTD, the STTD valuescorrects for basal endogenous losses, resulting in greater accuracy of estimating true digestibility. Shen et al. [213] estimated that basal endogenous losses of P in corn account for about 26% of the daily P requirement for pigs. Because ATTD values underestimate Ca and P digestibility, especially in feed ingredients and diets containing low concentrations of these minerals, STTD and TTTD values should be used instead of ATTD [33]. In addition, unlike ATTD values, STTD P values are additive for all feed ingredients, which is essential for use in diet formulation [214,215].

To optimize P utilization when feeding diets containing plant-based ingredients containing variable but relatively high concentrations of phytate to swine, exogenous phytase enzymes must be added to increase the proportion of dietary P used by the animal, reduce P excretion in manure, and minimize the antinutritional effects of phytate on digestibility of other nutrients [216,217,218]. Achieving complete hydrolysis of phytate to phosphate and inositol in swine diets is a complex process that requires dietary conditions which allow capturing the animal performance benefits resulting from phytate degradation. To achieve these benefits, complete enzymatic removal of all high molecular weight esters of phytic acid must occur through the use of exogenous enzymes. However, the effectiveness of this process is dependent on specific alterations in diet nutrient density, as well as the subsequent dephosphorylation of lower-molecular-weight esters to free phosphate and inositol [219]. Several interacting dietary and nutritional factors must be considered to achieve this goal, including: (1) phytate concentration, source, and solubility; (2) protein concentration and type; (3) phytase type and dose; (4) vitamin D status of the animal; (5) water quality characteristics; (6) dietary calcium concentration; and (7) the need for additional exogenous enzymes [219]. Phytase responses are complex and have been described usinga meta-analysis of published data [220]. Although responses to phytase supplementation in pig diets could be predicted, the accuracy of prediction was limited due to the large variation in P digestibility and digestible P concentration of pig diets in the data set [220]. While achieving “phytate-free” nutrition is possible, it will require strategic use of phytase in swine diets based on cross-validated animal models that can predict outcomes based on real-time analysis of phytate and dietary mineral, AAs, and energy balance [219].

## 3. Benefits and Limitations Using In Vitro and Ex Vivo Determinations of the Nutri-Physiological Value of Feed Ingredients

The use of in vivo models is generally considered to be more accurate for assessing the nutri-physiological properties of feed ingredients compared with in vitro and ex vivo methodologies. However, in vivo models have the disadvantages of being subjected to confounding factors such as uncontrolled environmental conditions, differences in microbiome and immune status of animals, and substantial variability in individual pig responses within the same dietary treatment. Because of these confounding factors, use of in vitro and ex vivo methodologies provide several advantages for specific applications when evaluating feed ingredients. First, they provide a faster and less-expensive means for dynamically estimating the nutri-physiological value of feed ingredients compared with expensive and cumbersome in vivo methods. In fact, some nutrition researchers have recognized that holistic and accurate evaluation of the nutritional value of feedstuffs often requires separate assessment of specific components of digestive processes and their end products at the specific sites where the nutrient digestion and absorption occurs along the gastrointestinal tract [221]. Furthermore, use of in vivo methods, especially those that involve using surgically modified animals, require specialized facilities, equipment, and technician training, and are labor-intensive. There are also increasing ethical concerns regarding the use of surgically modified animals and other invasive methods to collect samples and data in in vivo nutrition studies [222]. Although many international organizations provide guidelines and regulatory requirements for humane care and use of animals in scientific experiments, suitable alternative methodologies to in vivo experiments are needed to overcome some of the disadvantages from using in vivo trials and avoid potential unethical concerns [222]. In general, there are two categories of alternative methodologies that have been used to evaluate the nutritional value of feed ingredients for swine, which include in vitro (near-infrared reflectance spectroscopy—NIRS, closed enzymatic, pH-stat) and ex vivo (Ussing chambers and intestinal enteroids) methods.

### 3.1. In Vitro Methods

Several in vitro methods have been developed and used for estimating nutrient digestibility in feed ingredients during the past several decades [197]. Although the use of NIRS has primarily focused on determining total nutrient content of feed ingredients and complete feeds, there is considerable interest in applying this technology for estimating the digestibility and bioavailability of nutrients. Several other in vitro methods have also been developed and used to estimate nutrient digestibility in feedstuffs for pigs, including: (1) dialysis-cell, (2) colorimetric, (3) pH-drop and pH-stat, and (4) filtration methods [197]. Dialysis-cell methods have not been used extensively because of the high cost of dialysis tubes, but these procedures involve enzymatic digestion of protein accompanied by continuous removal of low-molecular-weight compounds by dialysis to prevent end products from digestion from inhibiting enzyme activities [197]. Colorimetric methods have been used mainly for estimating starch digestibility in processed feeds [197], and AA digestibility in DDGS [185,186], but color scores have been shown to be poorly associated with SID of lysine and other AA among corn DDGS sources [187]. The pH-drop [223] and pH-stat [224] methods are relatively simple analytical procedures that have been used to estimate protein quality in processed feed ingredients such as soybean meal and involve measuring the change in pH after enzymatic digestion. However, studies have shown that data obtained from these in vitro methods were poorly correlated with in vivo values for several types of feed ingredients [224,225]. Lastly, many different variations of filtration methods have been used to estimate the total tract and ileal digestible nutrients using one-, two-, or three-step incubations with enzymes in closed systems [197]. Because NIRS and filtration methods have been the most widely used and have greater applicability for more accurately estimating the nutri-physiological properties of feed ingredients compared with other methods, these procedures are discussed in greater detail in the following sections.

#### 3.1.1. Near-Infrared Reflectance Spectroscopy (NIRS)

Near-infrared reflectance spectroscopy is a rapid, physically nondestructive, and inexpensive technique that is rapidly being adopted in the global feed industry to accurately estimate nutrient content of feed ingredients. The principle of this technique involves measuring the reflectance spectrum in the near-infrared wavelength region of the sample and comparing these data with reference spectra of known samples to provide a quantitative estimate of analytes of interest. Calibration of NIRS equipment is based on using nutrient-composition data derived from standard chemical analysis procedures and various statistical techniques. The accuracy and precision of NIRS estimates are dependent on the robustness and size of the data set used to create the calibration model. Near-infrared reflectance spectroscopy calibrations have been developed for determining proximate analysis components [226,227,228,229,230], total and digestible AA content [231,232], and energy values [233,234,235,236] of feed ingredients and finished feeds for pigs. The use of NIRS technology to measure the bioavailability of nutrients in various feedstuffs is also being explored. For example, NIRS calibrations have been developed to measure reactive lysine and the extent of heat-damage in wheat distillers grains [237]. However, as for all types of analytical procedures, NIRS has several advantages and disadvantages compared with traditional chemical analysis techniques [238].

Advantages
(1)Rapid scanning of the samples (less than 1 min).(2)Only a small amount of sample is needed for analysis.(3)Low cost because no chemical reagents are needed and a single operator can analyze a large number of samples in a short period of time.(4)Results are highly reproducible.(5)Multiple analytes can be determined in one operation.(6)Minimal (drying and grinding) or no sample preparation is needed.(7)Equipment can easily be used in different environments (e.g., ingredient processing, grain harvest, laboratory, feed mill).(8)High accessibility for online data capture and storage.(9)Some optical probes allow analyzing samples in situ.(10)Equipment is portable.

Disadvantages
(1)It is a secondary method that requires the use of data derived from chemical analysis or in vivo studies as reference values.(2)A large number of samples with variable composition and data with large variation is required for accurate and robust calibrations.(3)Highly trained personnel are required for calibration and validation of the results.(4)Continuous maintenance and updating of the calibration database is required.(5)Changes in chemical structure of nutrients that occur during the digestion process cannot be predicted using NIRS technology.(6)High initial cost for purchasing NIRS instruments.

#### 3.1.2. Closed In Vitro Filtration Methods

Closed filtration or multi-enzymatic methods have been developed to mimic part or all of the in vivo gastrointestinal tract digestion process and are used to estimate the dry matter and nutrient digestibility of various ingredients by using single or multiple enzymes and collecting undigested residues. Świȩch [196] summarized the various one-, two-, or three-step sample incubations and the types of enzymes used in these closed systems (Table 12). Although the use of a single protease in a one-step procedure is relatively simple and may provide some useful information regarding the extent of heat-damaged protein in feed ingredients, it is inadequate for estimating true AA and nutrient digestibility.

Historically, various two-step multienzymatic incubation methods, including pepsin–pancreatin [243,244,245], pepsin–pronase [245], pepsin–trypsin [246] or pepsin–jejunal fluid [247] were developed and used to estimate CP digestibility or ileal digestibility of AA. The use of the Boisen and Fernandez [254] procedure has resulted in highly reproducible two-step in vitro analysis of protein and AA digestibility of feed ingredients. However, it is important to recognize that when using the two-step procedure, in vitro digestible CP data are greater than in vivo values because no endogenous nitrogen losses occur [254]. Interestingly, when endogenous nitrogen losses were estimated in 17 different feedstuffs and included in the in vitro equations, the results were highly correlated with in vivo ileal digestibility of protein and AA in pigs (R^2^ = 0.92; [254]). Unfortunately, further validation of the prediction equations using 48 feed mixtures with known in vivo digestibility values for AID of protein and AA resulted in poorer correlations than observed for individual feed ingredients, which suggests that further refinement of this assay is needed [254].

The two-step enzymatic method has also been applied to predict in vitro STTD of phosphorus (P) in poultry studies [255,256,257,258,259], but it can also be applied in swine in vitro models because the large intestine does not play a role in the digestion process of dietary phosphorus [260]. In fact, Zhu et al. [261] reported a high correlation (R^2^ = 0.91) between in vitro P digestibility and in vivo STTD of P among 13 sources of rendered animal protein meals for swine. These results suggest that the two-step in vitro P digestibility assay is capable of good prediction of in vivo P digestibility of animal protein byproducts fed to swine.

The three-step in vitro system is designed to estimate nutrient digestibility of the entire gastrointestinal tract of pigs, is highly repeatable, and has been extensively evaluated in several independent studies. The first two steps of this method involve consecutive incubations of feed ingredients with various enzymes that mimic digestion in the stomach, small intestine, and large intestine [248,249,250,251,252]. Each incubation step is conducted at the optimum pH, temperature, and time, and undigested residues are collected by filtration, defatted with ethanol and acetone, and analyzed for dry matter and nutrient content. The third step involves the measurement of substrate disappearance during fermentation, along with fermentation kinetics of substrates and short-chain fatty-acid production [262,263,264]. In vitro studies have been validated to simulate large-intestine fermentability using fresh fecal samples from pigs [262,265,266]. Development and use of automated gas-production systems has improved measurement during the fermentation portion of the assay [267,268,269], and no differences in fermentation kinetics and production of microbial metabolites were observed between manual and automatic in vitro fermentation recording systems using swine fecal inocula [268].

Despite these encouraging results, none of these in vitro methods completely simulate the complex biochemical and physiological events that occur during the in vivo processes of energy [270] and nutrient digestion [271,272]. There are several reasons for this. First, the effects of antinutritional factors are rarely mimicked in the in vitro system, but they can have significant effects on in vivo nutrient digestion. Second, the effect of DF on feed intake and transit time cannot be replicated with two- or three-step enzymatic in vitro systems. Third, effects of the gut microflora are difficult to simulate, but the microbiome can play a significant role in digestion and fermentation processes. Therefore, these effects may vary depending on the fecal sampling methods used, antinutritional factors present, and the DF content of feed ingredients being evaluated. Lastly, the assumption that all soluble fiber is digestible is not correct, because it may contain high amounts of small peptides commonly associated with heat-treated proteins, that may not be absorbed [221].

In general, data derived from these in vitro procedures do not match in vivo determinations for nutrient digestibility of feed ingredients. However, two- and three-step in vitro assays can provide relatively rapid and cost-effective analysis, and serve as reasonable initial screening methods for assessing the magnitude of digestibility and fermentability differences among different sources of ingredients [272]. Furthermore, although initial attempts have been disappointing [270], combining in vitro estimates with chemical composition may enhance the accuracy of using prediction equations to enable more dynamic estimation of the nutri-physiological value of feed ingredients.

### 3.2. Ex Vivo Methods

#### 3.2.1. Ussing Chambers

The Ussing chamber system was first developed and introduced in the 1950s and is used for diffusion- or electrophysiology-based measurements [273]. Ussing chambers have several applications in studies designed to determine ion transport in tissues, drug and protein absorption, and several pathophysiological process in animals [274,275]. However, in regard to nutrition, they have primarily been used to study intestinal permeability and intestinal-barrier function in weaned pigs [276]. Numerous pig trials have been conducted to evaluate the effect of dietary interventions on the intestinal barrier function and absorption using Ussing chambers (Table 13). Several types of probe markers have been used, but most studies have used mannitol to assess intestinal barrier function. In addition, small intestine transcellular absorption can be determined using sodium-dependent glucose or glutamine to estimate the active transport over the intestinal epithelium in Ussing chambers [277].

The major advantages of using Ussing chambers are to study regional (duodenum, jejunum, ileum, or colon) permeability and barrier function. However, the main technical challenge of conducting experiments with Ussing chambers is the limited amount of time (up to 2.5 h) of maintaining viable intestinal tissue, which is not enough for the investigation of the reversible disruption of tight junctions and extensive metabolism determinations [278,279]. Therefore, use of this technique requires specialized and trained experts to conduct measurements and interpret results. Because the intestinal epithelium in the chamber is maintained under stable conditions, the lack of accounting for changes in hormonal, inflammatory, or other metabolic signals in the animal can lead to difficulty in interpreting results. Furthermore, using this assay usually involves exposing tissues to purified nutrient solutions that do not represent actual mixtures of nutrients provided to the intestine under normal in vivo conditions. In summary, experiments with Ussing chambers are very informative for assessing intestinal permeability of pigs but are of no use for determining nutrient digestion or absorption from a complex diet in an animal. For this reason, other ex vivo methods such as the use of cell cultures have become popular alternatives for studying gut nutri-physiology.

**Table 13 animals-11-01259-t013:** Summary of results from ex vivo studies to evaluate intestinal permeability of nursery pigs with Ussing chambers.

References	Age	Intestinal Segment	Probes (Markers) ^1^						Findings
			HRP	Man	GlySar	Na-Flu	Na^+^-Gluc	Na^+^-Glut	
[280]	26	Jejunum	X	X	X	X			Postweaning feed intake level did not change gut permeability in nursery pigs
[281]	28	Jejunum	X			X			Intestinal molecular permeability was not affected by the age of weaning and creep feeding
[282]	28	Jejunum					X		Dietary modification from milk- to grain-based sources did not affect HRP fluxes in nursery pigs
[283]	25	Jejunum	X			X			Intestinal macromolecular permeability was not affected by supplemental dietary tryptophan
[284]	26	Jejunum		X	X				Intestinal permeability was not different between piglets fed a high-lactulose and low-protein (HL/LP) diet compared with piglets fed control (milk-based) diet
[285]	26	Jejunum		X	X				Feeding dry pellets elevated transcellular permeability compared with wet feeding
		Jejunum		X	X				
[286,287]	28	jejunum		X			X	X	Paracellular permeability was not affected by supplementation of various probiotics (*E. farcium* and *B. cereus* var. *toyoi*)
[288]	24	Ileum, Colon		X			X		Feeding diets containing 2.5% and 5% spray-dried porcine plasma reduced ileal permeability of pigs on day 7 postweaning
[289]	28	Ileum		X			X		Increasing dietary Zn level from 100 to 2500 ppm at weaning increased intestinal permeability and reduced diarrhea
[290]	28	Jejunum		X			X		Dietary copper disturbed intestinal-barrier function by increasing transepithelial conductance
[291]	7	Ileum		X			X		Intestinal permeability increased by 89% in the ileum of piglets deficient in dietary threonine (6.5 g/kg) compared with piglets fed the control diet containing 9.3 g/kg threonine
[292]	14–17	Jejunum						X	Long-chain (n-3) PUFA supplementation of maternal diets had no effect on total or passive ion transport of their progeny
[293]	15–19	Jejunum					X		Feeding maternal diets containing long-chain (n−3) PUFA resulted in upregulated glucose flux in piglet jejunum

^1^ HRP = horseradish peroxidase; Man = mannitol; GlySar = glycylsarcosine; Na-Flu = sodium-fluorescein isothiocyanate; Na^+^-gluc = sodium-dependent glucose; Na^+^-glut = sodium-dependent glutamine.

#### 3.2.2. Enteroids

Use of intestinal cell cultures offers many potential benefits for studying specific and complex mechanisms involving the chemical and nutritional composition of feed ingredients associated with physiological responses of pigs. Historically, two-dimensional intestinal cell line models, such as Caco-2 [294], HT-29 [295,296], HCT116 [297], and SW480 [298], have been used in nutritional studies. These cell cultures often consist of a single cell type and are grown in a monolayer, which provides a simple two-dimensional structure for conducting experiments and interpreting outcomes, but they are incapable of providing information for understanding complex relationships between diverse cell types and morphological structures of the small intestine. Furthermore, two-dimensional cell-culture systems do not represent the complex three-dimensional structures of intestinal epithelial cells and can easily mutate during establishment [299,300]. Therefore, to overcome the limitations of two-dimensional cell cultures, the use of enteroids has become an emerging ex vivo approach for evaluating the complex three-dimensional structures of intestinal epithelial cells for swine nutrition [301,302] and pathogen studies [303,304].

Enteroids are three-dimensional structures that originate from embryonic stem cells, induced pluripotent cells, or adult stem cells from intestinal tissue, and are grown in cell culture. Unlike two-dimensional cell lines, enteroids have all of the differentiated cells found in the small intestine, including lumen, villi and crypts of enterocytes, enteroendocrine cells, goblet cells, tuft cells, Paneth cells, and stem cells [305]. Therefore, their structure and hierarchy highly resemble the in vivo intestinal epithelium.

Compared with other in vitro models, use of enteroids provides the opportunity to study the effects of diet and nutrients on intestinal growth and development, ion and nutrient transport, secretory and absorptive functions, intestinal barrier, cell differentiation, intestinal disease, gene expression, and other responses of interest in the gastrointestinal tract [301,302,303,304,305]. In addition, enteroids: (1) possess most of the cell types of intestinal epithelium; (2) can be rapidly established from adult stem cells and pluripotent stem cells; (3) are stable in long-term cultures (at least 1.5 years) and do not show genetic or physiological changes; (4) require only a single intestine donor, which reduces the number of animals needed for experimentation; (5) cultures can be multiplied and maintained for several months in the laboratory once they are developed; and (6) have minimal ethical issues and costs compared with using invasive in vivo methods. However, there are also several disadvantages for using enteroids in nutritional experiments. First, there are no standard protocols and guidance for establishing cell cultures of enteroids. Second, establishing and maintaining enteroid cultures may be more expensive compared with conventional cell lines. Lastly, the potential variation between individual stem-cell donors and protocols may cause misinterpretation of experimental results.

Knowledge of and interest in using enteroid-based technologies continues to increase relative to their many advantages for animal nutrition research. Although limited studies have used enteroid models in swine-nutrition research, enteroids represent a promising ex vivo technology that could greatly enhance our knowledge of nutri-physiological characteristics of feed ingredients for swine in the future.

## 4. Nutrient Digestion Kinetics of Feed Ingredients

Traditional feed ingredient evaluation models have described nutrient digestion, absorption, and metabolism as sequential events in which nutrients are made available for growth in a way that is quantified by the total amount of nutrient digested and absorbed in the gastrointestinal tract [63]. For example, the disappearance of AAs in the small intestine of growing pigs has been correlated to the lean deposition rate of growing pigs [306]. Although the use of SID of AAs is considered to be the “gold standard” for estimating the proportion of total AAs in feed ingredients and diets that will be utilized to meet the AA requirements of pigs [63], there continues to be a need for further improving the efficiency of nutrient utilization in pig production, especially for nitrogen. Approximately 54% of nitrogen supplied in pig diets is not utilized for growth or productive purposes and is instead excreted in feces and urine [307]. Therefore, it is necessary to study the reasons for the digestive and metabolic inefficiency of utilizing AAs and other required nutrients in swine diets.

The inefficiency of nitrogen utilization may be attributed to the concept that protein deposition is considered as an “all or nothing” event, in which all AAs must be present at the exact moment of protein synthesis or they will not be utilized [308]. Assuming this concept is true, then synchronizing the timing and rate of AA digestion, absorption, and appearance in systemic circulation needs to coincide with energy availability for protein synthesis to improve the efficiency of nitrogen utilization in growing pigs. Therefore, the impact of dynamic digestion processes, their coordination, and interaction with other feed ingredients in the diet needs to be considered when evaluating the nutri-physiological value of feed ingredients for growing pigs.

The rate and amount of AA disappearance along the small intestine of pigs varies among feed ingredients (Figure 1), where AA disappearance in the most proximal section of the small intestine is greatest for dried porcine plasma and is linear over time compared with soybean meal and rapeseed meal, while AA disappearance in wheat gluten occurs as a quadratic response over time [309]. Specifically, there is rapid appearance of AAs in systemic blood circulation in pigs consuming diets containing wheat gluten before reaching a plateau and declining (Figure 1). For pigs consuming diets containing dried porcine plasma, the rate of AA appearance in systemic blood was initially slower than that observed for pigs fed wheat gluten, but continued to increase linearly over time even after AA concentration was decreasing in pigs fed wheat gluten (Figure 1). Compared with wheat gluten and dried porcine plasma, pigs fed soybean meal and rapeseed meal had a slower and more stable rate of AA appearance in blood (Figure 1). Therefore, feed ingredients with significant contributions of dietary AAs can be classified as slow, intermediate, and rapid digestible sources of AAs. Wheat gluten and dried porcine plasma are considered rapid digestible sources of protein, while soybean meal and rapeseed meal are classified as slow sources of digestible protein (Figure 2). Because the in vivo kinetics of protein digestion in the small intestine of pigs is complex and difficult to measure, in vitro digestibility assays have been used as an inexpensive and simpler alternative [309,310]. However, there are no publicly available databases or sources of information for digestion kinetics data of feed ingredients that can be used in routine diet formulation. Therefore, future feed ingredient evaluation studies should focus on characterizing the digestion kinetics of common feed ingredients, which can be used in dynamic and mechanistic mathematical models that include digesta transit time, nutrient hydrolysis, and absorption rates to predict nutri-physiological responses from mixtures of various types of feed ingredients [310].

Digestion and absorption of simple carbohydrates, such as glucose and sucrose, require less time than that of more complex carbohydrates such as corn starch [310]. Differences in the rate of digestion, absorption, and appearance of glucose in portal vein circulation are also observed among different chemical forms of starch, and these differences in kinetics can be classified as slow, rapid, and resistant starch [311]. In vitro measurement of the release of starch coupled with the predicted gastric emptying rate allows for the rapid determination of in vivo peak of blood glucose appearance in the postprandial portal blood of pigs [312]. Resistant starch provides about 83% of the energy value of digestible starch and can be quantified using tracer and calorimetric techniques [313]. These methods to estimate the true energy value of different chemical forms of starch are necessary because there are differences in digestibility of starch among cereal grains. The total disappearance of starch in the small intestine of pigs can be a low as 74% in barley or as high as 98% in corn [314,315]. While these static values of starch digestibility are currently being used for the determination of the energy value of feed ingredients [316], there is increasing evidence that these static values are insufficient to describe the effects on growth performance of pigs when the kinetics of fiber degradation are considered.

Digestion and absorption kinetics of starch and protein rich ingredients have been the most studied and documented relative to pig growth performance and health, but these characteristics are less defined and described for high-fiber feed ingredients. There are, however, clear differences among feed ingredients in the site and rate of degradation of fiber in the gastrointestinal tract of pigs [317]. The traditional definition of DF refers to the portion of dietary carbohydrates that are indigestible in the small intestine of pigs [318]. While it is true that fiber degradation in the small intestine may not occur from endogenous enzyme activity in the small intestine, it is clear that the extent of disappearance of DF in the small intestine of pigs varies among different types of high-fiber feed ingredients (Figure 3).

For example, in pigs consuming diets containing corn DDGS, which contains high concentrations of insoluble fiber, about 20.6% of the fiber disappears in the small intestine within about 8–12 h (Figure 3). Therefore, this portion of DF in DDGS can be regarded as fast or readily degradable fiber. The second DF fraction is more resistant to degradation and requires about 24–36 h to degrade another 28.9% of the total DF in DDGS. This form of DF degradation occurs in the large intestine as a result of the action of microbial enzymes and fermentation, and is generally characterized as fermentable or slow-degradable fiber. Finally, the remaining 50.5% of total DF in DDGS is not degraded in the intestinal tract of pigs but is excreted in feces. This portion of fiber is regarded as recalcitrant or resistant to degradation.

Determination of kinetics of DF degradation using three-step in vitro digestibility systems has begun to provide insight regarding the rate and type of DF degradation in common high-fiber feed ingredients. In vitro, fiber degradation may occur by fiber solubilization in the stomach (step 1) and small intestine (step 2) during the pepsin and pancreatin digestion phase, in which pepsin degradation requires 2 h duration, while pancreatin degradation lasts for 4 h. Additional DF disappearance from fermentation is measured using fecal inoculum in step 3. Studies have also been conducted to evaluate the use of nonstarch polysaccharide (NSP) degrading enzymes, which have shown increased disappearance of DF in wheat middlings and corn DDGS in the small intestinal phase of digestion [322]. However, the use of NSP enzymes during the fermentation phase decreased the gas production in samples that were predigested with NSP enzymes. Therefore, results from this study suggests that use of NSP enzymes shifts degradation of fiber from the large intestine, which is the traditional site of degradation, to degradation in the small intestine [322]. These in vitro observations are consistent with those from in vivo studies showing that disappearance of fiber in the duodenum of growing pigs is greater when pigs consumed diets that contained NSP enzymes than when no enzymes were added to the diets [323]. In fact, 7.4% of DF disappeared in the duodenum of pigs fed corn DDGS and NSP enzymes, compared with 1.4% of DF disappearance in pigs fed corn DDGS with no NSP enzymes. However, the apparent total tract disappearance of DF in pigs fed corn DDGS diets with and without NSP enzymes was similar (67.9% vs. 67.7%, respectively). Therefore, there is increasing evidence to suggest that a portion of DF in feed ingredients is composed of heterogenous carbohydrates that can be degraded rapidly in the duodenum of growing pigs. While the goal of adding NSP enzymes to swine diets is to increase the digestibility of DF, these enzymes appear to alter the site along the gastrointestinal tract where DF is degraded, and has also been shown to affect mucin gene expression and the immune response profile of growing pigs [324].

Several studies have been conducted to evaluate the application of using kinetics of protein, starch, and fiber degradation of feed ingredients in diet formulations to improve growth performance and gut health of nursery pigs [325,326,327]. Results from these studies have shown an 18% improvement in growth rate when weaned pigs were fed diets consisting of potato and soybean protein concentrate compared with pigs fed diets with wheat-gluten protein [325]. However, diarrhea incidence in pigs fed diets containing potato and soybean protein concentrate was greater (38%) than in pigs fed wheat-gluten diets (27%), but when potato and soybean protein concentrate was combined with resistant starch, the incidence of diarrhea decreased. In another study, pigs fed diets composed of rapidly digestible protein and resistant fiber had 9% greater feed intake and 27% greater body weight gain than pigs fed conventional diets. Interestingly, the impact of a severe gut health challenge was less in pigs fed diets with fast digestible protein than pigs fed conventional diets that were not supplemented with pharmacological levels of zinc oxide. Therefore, it appears that selection of dietary protein sources based on the kinetics of digestion may be a practical alternative to adding supranutritional concentrations of zinc from zinc oxide in diets for weaned pigs [326].

When starch is degraded and absorbed in the small intestine, it serves as the primary source of energy in diets for pigs, and it is also a potential source of energy for microbes in the large intestine. Providing energy and nutrients to beneficial microflora in the large intestine is considered necessary for decreasing postweaning diarrhea and maintaining gut health in pigs. Consequently, understanding the kinetics of starch digestion may also be beneficial for enhancing growth performance of pigs. The kinetics of starch digestion has been measured by the amylase diffusion rate, and it appears that formulating diets with sources of starch that have high intrinsic digestibility increases growth performance of nursery pigs [328]. This enhanced growth performance may be due to the greater digestibility of starch in feed ingredient sources that have greater intrinsic amylase diffusion rates. Although grain particle size has been shown to be the major determinant of the amylase diffusion rate, results from a recent study suggests that milling grain to particle sizes less than 600 microns (commonly usually used in the feed industry) and steam-flaking increases the amylase diffusion rate of some grains such as sorghum [328].

In summary, the rate and sites of protein, starch, and DF degradation along the gastrointestinal tract differs among feed ingredient sources, and the kinetics of degradation can be measured using in vitro and in vivo models. Although the use of protein, starch, and DF degradation kinetics in practical diet formulation is still at the nascent stage, results from using this approach are promising for improving pig growth performance and health. Many of the current nutrient digestion kinetics models have only been applied to evaluating individual feed ingredients, and do not account for the significant interactions between protein and DF, as well as with other dietary nutrients in complete feeds. Characterizing these interactions will require more extensive modeling to accurately represent the complex and often nonlinear interactions among dietary components. Finally, databases that provide nutrient degradation kinetics data need to be developed to apply this new approach of feed ingredient evaluation in routine practical swine diet formulations.

## 5. Functional Ingredients and Nutrients

Functional foods have been defined as those that provide a health benefit beyond satisfying nutrient requirements in ways not anticipated by traditional nutrition science [329]. Reviews have been published that describe the complex interactions between nutrition, immunology, gastrointestinal physiology, and the gut microbiome in human nutrition and health [330,331]. This definition can also be directly applied to defining functional feed ingredients for animals. In fact, practical nutrition approaches for improving intestinal immunity during the weaning transition for pigs have been reviewed [332]. Although feed ingredients can be described as complex physical and chemical matrices that provide energy and various nutrients to animal diets to support body maintenance, growth, and reproductive functions, many ingredients are comprised of non-nutritional, bioactive compounds or properties that promote animal health. Health-promoting characteristics of functional feed ingredients may include one or more of the following: (1) antimicrobial properties, (2) antioxidant properties, (3) an influence on gut microbiome composition and function, (4) a reduction inflammation by enhancing immune signaling and responses, (5) stimulation of feed consumption, and (6) improvement of gut health. However, it is difficult to distinguish between functional feed ingredients and functional nutrients because ingredients contain combinations of functional nutrients or chemical constituents that provide specific health-promoting effects. Therefore, functional ingredients and nutrients collectively provide positive effects on digestive processes and the gut microbiome, as well as gastrointestinal and systemic health.

Numerous reviews have been published to describe the specific functional ingredients and nutrients along with their beneficial health effects when included in swine diets (Table 14). The presence or absence of functional components in feed ingredients should be considered when evaluating their economic value, which should include potential impact on reduced morbidity, mortality, and medication costs, as well as the necessity and cost of adding growth- and health-promoting feed additives to the diet. Furthermore, understanding the functional properties of ingredients and nutrients can improve the likelihood and potential magnitude of positive growth and health responses, while avoiding antagonistic effects when selecting and adding various feed additives to swine diets.

One of the most interesting but underexplored health promoting components of feed ingredients are the many naturally occurring polyphenolic compounds in feed ingredients. Several phenolic compounds are associated with DF in many plant-based foods and feeds, which have potent antioxidant and reactive oxygen species scavenging properties that provide protection against oxidative damage [369]. However, these functional attributes have generally been considered as separate components of DF because they differ in molecular structures, physicochemical and biological properties, and metabolism [370]. Several studies have shown that polyphenolic compounds provide beneficial antioxidant, bactericidal, and immunostimulatory activities in animal nutrition [349,350,351,352,371]. Hydroxycinnamic acids are potent antioxidants that are the most abundant phenolic compounds associated with DF in cereal grains, and are predominately in the form of ferulic acid, with lesser amounts of diferulic, sinapic, *p*-coumaric, and caffeic acids [372]. In fact, about 95% of phenolic compounds present in cereal grains are linked to polysaccharides, which are primarily diferulates covalently bound by ester linkages to α-arabinoxylans [372]. Although some sources of DF have been characterized as having very high antioxidant capacity that contributes to their functional properties [373], their relative effectiveness in improving oxidative status when fed to animals is controversial because the bioavailability of phenolic compounds varies among DF sources [374]. Based on data summarized by Vitaglione et al. [372], phenolic acids are more concentrated in the bran fraction than in whole grains (Table 15). Furthermore, although concentrations are highly variable, ferulic acid and *p*-coumeric acid content appear to be found in the greatest concentrations in corn grain and bran compared with other grains and their respective bran fractions (Table 15).

Vitamins also play an essential role in the health and immune functions of animals [375,376] and ensuring their adequacy under modern commercial pork production conditions with various oxidative and pathogen stressors can be a challenge. Vitamin deficiencies have been shown to increase susceptibility of animals to infectious enteric diseases [377], including inflammatory diseases of the gastrointestinal tract [378,379]. Although several studies have been conducted to determine the role of vitamins on gut microbiome, function, health, and disease prevention in humans, limited studies have been conducted to evaluate these effects in food-producing animals [368]. However, as summarized in Table 16, several vitamins are directly associated with inhibiting enteric infections in pigs [321]. Therefore, future studies are needed to re-evaluate vitamin requirements of pigs from modern genetic lines and under various disease and environmental challenge conditions commonly found in commercial pork-production systems. Without this essential information, there is minimal guidance on recommended dietary vitamin supplementation levels necessary for optimizing swine health.

## 6. Conclusions

Numerous methods and measures have been used to estimate nutritional and physiological responses of pigs based on chemical composition data when feeding diets containing various mixtures of feed ingredients. However, precision swine nutrition requires the use of the most accurate measures and multiple criteria to predict nutri-physiological responses of various types of feed ingredients. The use of traditional proximate analysis measures (i.e., crude protein, crude fat, crude fiber) should be avoided because they provide very limited useful information about the nutritional and physiological properties of feed ingredients for swine. Theuse of the net energy system, SID AA content, and STTD phosphorus content measures are encouraged for evaluating feed ingredients and formulating swine diets, but additional measures should be considered in feed ingredient databases, including: (1) water activity; (2) solubility, fermentability, viscosity, and prebiotic effects of dietary fiber; (3) lipid and protein oxidation measures; and (4) estimates of true bioavailability of amino acids, vitamins, and minerals. Furthermore, assessing the digestion kinetics of various dietary components (i.e., starch, dietary fiber, protein) is a promising new approach that allows matching the various rates and quantities of nutrients digested in mixtures of diverse feed ingredients to optimize nutri-physiological responses in swine. Combining net energy and digestible nutrient prediction equations derived from meta-analyses of large and robust data sets with NIRS determinations of total nutrient content of feed ingredients may be the most practical and accurate approach for dynamic estimation of nutrient loading values for feed formulation. Use of two- and three-step in vitro digestibility systems, intestinal enteroids, and LC-MS platforms to characterize digestive and metabolic responses will be necessary for determining nutrient bioavailability of micronutrients and functional properties of various ingredients. Perhaps the biggest challenge for the future is to develop an effective systems biology approach for integrating large, complex data sets involving numerous nutritional and physiological response criteria so thathighly accurate, dynamic, mechanistic, and predictive mathematical models can be developed and implemented for enhancing precision swine nutrition.

## Figures and Tables

**Figure 1 animals-11-01259-f001:**
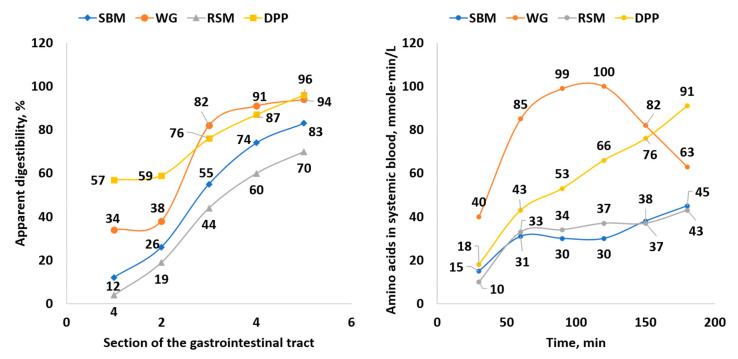
Apparent digestibility of crude protein along sections of the small intestine (1–4), total tract digestibility (5; left panel), and AA concentration in systemic blood at different time points (right panel) in pigs fed diets containing soybean meal (SBM), wheat gluten (WG), rapeseed meal (RSM), and dried porcine plasma (DPP) (adapted from [309]).

**Figure 2 animals-11-01259-f002:**
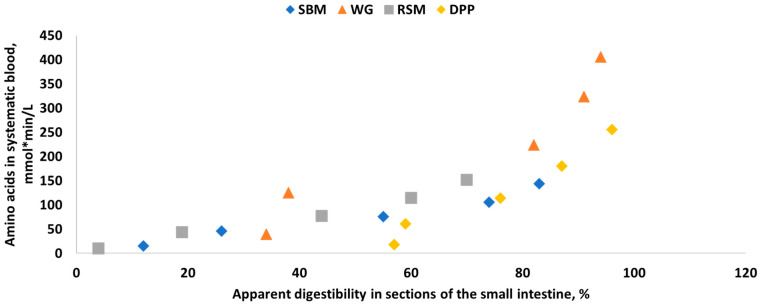
Relationship between apparent digestibility of CP along sections of the small intestine and the postprandial appearance of AA in systemic blood of pigs fed soybean meal (SBM), wheat gluten (WG), rapeseed meal (RSM), and dried porcine plasma (DPP) (adapted from [309]).

**Figure 3 animals-11-01259-f003:**
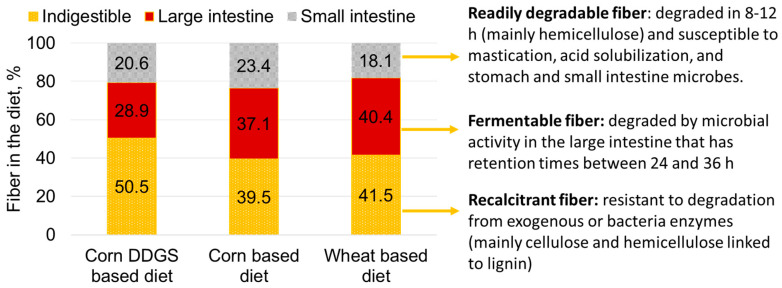
Total dietary fiber fractions degraded in the small intestine and large intestine, and the indigestible portion of diets with corn distillers dried grains with solubles (DDGS), and corn- and wheat-based diets (adapted from [319,320,321]).

**Table 1 animals-11-01259-t001:** Moisture content of several grains relative to water activity at 25 °C (adapted from [55]).

Grain	Moisture Content (%) at Various a_w_			
Water activity, a_w_	0.60	0.65	0.70	0.75
Rice	13.2	13.8	14.2	15.0
Oats	11.2	12.2	13.0	14.0
Rye	12.2	12.8	13.6	14.6
Barley	12.2	13.0	14.0	15.0
Corn	12.8	13.4	14.2	15.2
Sorghum	12.0	13.0	13.8	14.8
Wheat	13.0	13.6	14.6	15.8

**Table 2 animals-11-01259-t002:** Water activity and content in fine and coarse milled feed ingredients (adapted from [54]).

Ingredient	Mean Particle Size (μm)		Water Activity, a_w_		Water Content, %	
	Coarse	Fine	Coarse	Fine	Coarse	Fine
Soybean meal	1430	342	0.657	0.596	11.8	11.4
Barley	2423	314	0.638	0.496	13.0	11.0
Rapeseed cake	708	310	0.502	0.481	8.7	8.2
Corn ^1^	-	-	0.605	0.564	12.4	11.9

^1^ No particle size data were provided.

**Table 3 animals-11-01259-t003:** Summary of carbohydrate fractions, ingredient sources, and physiological properties of nonstarch polysaccharides in foods and feed ingredients (adapted from [94]).

Characteristic	Nonstarch Polysaccharides									
	Cellulose	Hemicelluloses	β-glucans	Pectins	Gums and Mucilages	Fructans				
Fractions	-	Arabinogalactans Glucans Arabinoxylans Glucuronoxylans Xyloglucans Galactomannans Pectin substances	Oat β-glucan, Barley β-glucan	-	Galactomannans Guar (PHGG) Locust bean gum (carob) Carob galactomannan Tara galactomannan	Gum (acacia) Gum (karaya) Gum (tragacanth)	Alginates Agar Carrageenan	Xanthan Gellan	Psyllium	Inulin Oligofructose
Ingredient sources	Brans Legumes Nuts Peas Cereals Functional fibers	Brans Cereal grains Legumes Nuts Vegetables Fruit Functional fibers	Oats Barley Rye	Fruits Vegetables Legumes Potato Sugar beets	Legumes Seed extracts (endosperm)	Tree extracts	Seaweed extracts (algal polysaccharides)	Microbial gums	Outer layer of seeds of plantain family	Chicory root Onion Artichoke Agave Wheat
Physicochemical properties ^a,b,c,d^	Insoluble, Poorly to moderately fermentable	Insoluble, Poorly to moderately fermentable	Soluble, Viscous, Readily fermentable	Soluble, Readily fermentable	Soluble, Viscous (some), Readily fermentable	Soluble, Viscous (some gums), Readily fermentable	Soluble, Viscous (some), Readily fermentable	Soluble, Viscous (some), Readily fermentable	Soluble, Highly viscous, Not fermentable	Soluble, Nonviscous, Fermentable

^a^ Soluble fiber has the ability to dissolve in water. ^b^ Insoluble fiber does not dissolve in water and remains as discrete particles. ^c^ Viscosity is the ability of some polysaccharides to thicken and form a gel when hydrated. ^d^ Fermentability is the extent of fiber that resisted digestion in the small intestine be degraded by microbiota in the cecum and colon to produce short-chain fatty acids and gas.

**Table 4 animals-11-01259-t004:** Summary of carbohydrate fractions, ingredient sources, and physiological properties of nondigestible oligosaccharides and other fibers found in plants and animals (adapted from [94]).

Characteristic	Nondigestible Oligosaccharides (Short-Chain Oligosaccharides)	Other Fibers Found in Plants				Fibers Found in Animals (Fungi, Yeast, and Invertebrates)	
		Lignin	Cutin	Suberin	Waxes	Chitin	Chitosan (Commercially Produced from Chitin)
Fractions	Fructooligosaccharides (FOS)/Neosugar Galactooligosaccharides Xylooligosaccharides Arabinoxylanoligosaccharides (AXOS) Soybean oligosaccharides	-	-	-	-	-	-
Ingredient sources	-	Woody plants or outer layer of cereal grains	-	-	-	-	-
Physicochemical properties ^a,b,c,d^	Soluble, Nonviscous, Readily/rapidly fermentable	Insoluble, Poorly fermentable	Insoluble, Poorly fermentable	Insoluble, Poorly fermentable	Insoluble, Poorly fermentable	Insoluble, Poorly fermentable	Insoluble, Poorly fermentable

^a^ Soluble fiber has the ability to dissolve in water. ^b^ Insoluble fiber does not dissolve in water and remains as discrete particles. ^c^ Viscosity is the ability of some polysaccharides to thicken and form a gel when hydrated. ^d^ Fermentability is the extent of fiber that resisted digestion in the small intestine be degraded by microbiota in the cecum and colon to produce short-chain fatty acids and gas.

**Table 5 animals-11-01259-t005:** Summary of carbohydrate fractions, ingredient sources, and physiological properties of resistant starch and chemically synthesized carbohydrate compounds (adapted from [94]).

Characteristic	Resistant Starch			Chemically Synthesized Carbohydrate Compounds				
	Physically Inaccessible Starch	Native Starch Granules	Retrograded Starch	Chemically Modified Resistant Starch	Dextrin	Polydextrose (e.g., Litesse™)	Cellulose-Derived	PolyGlycopleX (PGX)
Fractions	-	High amylose starch High amylose maize resistant starch type 2 High-maize 260	-	-	Resistant maltodextrin ^1^ Fibersol Pine fiber Wheat dextrin Nutriose™ Corn fiber	-	Methylcellulose Hydroxypropylmethylcellulose Hydroxyethylmethylcellulose Ethylmethylcellulose/cellulose gum	-
Ingredient sources	Legumes Seeds Whole grains	Unripe banana Raw potato (ungelatinized starch granules)	Heated and cooled starch	Modified food starch	Hydrolyzed food starch	-	-	-
Physicochemical properties ^a,b,c,d^	Poorly soluble, Variable fermentability	Poorly soluble, Variable fermentability	Poorly soluble, Variable fermentability	Soluble, Nonviscous, Readily fermentable	Viscous, Variable fermentability	Nonviscous, Readily fermentable	Viscous, Not fermentable Not gel-forming	-

^1^ Also referred to as maltodextrin, indigestible dextrin, resistant dextrin, and pyrodextrin. ^a^ Soluble fiber has the ability to dissolve in water. ^b^ Insoluble fiber does not dissolve in water and remains as discrete particles. ^c^ Viscosity is the ability of some polysaccharides to thicken and form a gel when hydrated. ^d^ Fermentability is the extent of fiber that resisted digestion in the small intestine be degraded by microbiota in the cecum and colon to produce short-chain fatty acids and gas.

**Table 6 animals-11-01259-t006:** Summary of published estimates of digestible energy (DE) and metabolizable energy (ME) content of soybean oil in swine diets.

Pig Body Weight	Diet Inclusion Rate	DE, kcal/kg	ME, kcal/kg	Reference
13 kg	5%	8993–9038	8813–8856	[141]
50 kg	5%	8181–9049	8017–8868	[141]
38 kg	4, 6, 8, 10%	4% = 8243	4% = 7966	[142]
		6% = 8419	6% = 8190	
		10% = 8911	8% = 8422	
		8% = 8775	10% = 8797	
34 kg	5 and 10% using two different basal diets	Corn–soybean meal	Corn–soybean meal	[143]
		5% = 8357	5% = 8099	
		10% = 8410	10% = 8854	
		Corn starch casein	Corn starch casein	
		5% = 8054	5% = 7896	
		10% = 8410	10% = 8319	
19 kg	7.13%	9979	-	[144]
10 kg	6.7%	8567	8469	[145]
15 kg	10%	8315	8368	[146]
-	-	8749	8574	[63]
-	-	7977	7906	[147]
-	-	8600	8300	[148]

**Table 7 animals-11-01259-t007:** Summary of estimates of net energy (NE) content of soybean oil in swine diets.

Pig Body Weight	Diet Inclusion Rate	NE, kcal/kg	Reference
22 kg	5 or 10%	5% = 4561 10% = 4781	[136]
84 kg	5 or 10%	5% = 5585 10% = 4578	[136]
31 kg	5 or 10%	5% = 7989 10% = 8132	[149]
13 kg	5%	7756–7795	[141]
50 kg	5%	7055–7804	[141]
-	-	7545	[63]
-	-	7117	[147]
-	-	7364	[148]

**Table 8 animals-11-01259-t008:** Summary of published digestible energy (DE), metabolizable energy (ME), and net energy (NE) prediction equations for lipids fed to swine.

Pig Body Weight	Equation ^1^	R ^2^	Reference
19 kg	DE (kcal/kg) = 10,267 − (110.3 × FFA, %) − (41.8 × C16:0, %) − (39.7 × C18:0, %) − (98.0 × U:S) + (6.4 × iodine value)	0.97	[144] ^2^
13 kg	DE (Mcal/kg) = 9.363 − (0.097 × FFA, %) − (0.016 × n-6:n-3) − (1.24 × C20:0, %) − (5.054 × insoluble impurities, %) + (0.014 × C16:0, %)	0.81	[141] ^3^
	ME (Mcal/kg) = 9.176 − (0.095 × FFA, %) − (0.016 × n-6:n-3) − (1.215 × C20:0, %) − (4.953 × insoluble impurities, %) + (0.014 × C16:0, %)	0.81	[141]
	NE (Mcal/kg) = 8.075 − (0.093 × FFA, %) − (0.014 × n-6:n-3) − (1.07 × C20:0, %) − (4.359 × insoluble impurities, %) + (0.013 × C16:0, %)	0.81	[141]
	DE (kcal/kg) = 37.89 – (0.0051 × FFA, g/kg) – 8.20^(−0.515 × U:S)^/0.004184	-	[129]
50 kg	DE (Mcal/kg) = 8.357 + (0.189 × U:S) − (0.195 × FFA, %) − (6.768 × C22:0, %) + (0.024 × PUFA, %)	0.81	[141]
	ME (Mcal/kg) = 8.19 + (0.185 × U:S) − (0.191 × FFA, %) − (6.633 × C22:0, %) + (0.023 × PUFA, %)	0.81	[141]
	NE (Mcal/kg) = 7.207 + (0.163 × U:S) − (0.168 × FFA, %) − (5.836 × C22:0, %) + (0.021 × PUFA, %)	0.81	[141]
	DE (kcal/kg) = 36.898 − (0.0046 × FFA, g/kg) − 7.33^(−0.906 × U:S)^/0.004184		[129]
Lactating sows	DE (kcal/kg) = 8381 − (80.6 × FFA, %) + (0.4 × FFA ^2^, %) + (248.8 × U:S) − (28.1 × U:S ^2^) + (12.8 × FFA, % × U:S)	0.74	[133] ^4^

^1^ FFA = free fatty acids; U:S = unsaturated to saturated fatty acid ratio; n-6:n-3 = omega 6 to omega 3 fatty acid ratio; PUFA = polyunsaturated fatty acids. ^2^ Equation derived from determining DE content and chemical composition of butter fat, canola oil, coconut oil, fish oil, flaxseed oil, lard, olive oil, palm oil, soybean oil, and tallow. ^3^ Equations derived from determining DE, ME, and NE content and chemical composition of an animal–vegetable blend, canola oil, two sources of choice white grease, coconut oil, two sources of corn oil, fish oil, flax oil, palm oil, poultry fat, two sources of soybean oil, and tallow. ^4^ Equation derived from determining DE and chemical composition of choice white grease, choice white grease acid oil, soybean oil, soy–cotton acid oil, and animal–vegetable blend.

**Table 9 animals-11-01259-t009:** Comparison of in vivo determined digestible energy (DE) content in distillers corn oil with variable free fatty acid content and predicted DE content using equations from [151] for young pigs (adapted from [150]).

Criterion	Corn Oil Composition				
Free fatty acids %	0.04	4.9	12.8	13.9	93.8
UFA:SFA ^1^	6.13	5.00	5.61	5.00	4.81
DE actual, kcal/kg	8814	8828	8036	8465	8921
DE predicted, kcal/kg	8972	8848	8794	8741	7775

^1^ UFA:SFA = unsaturated fatty acid to saturated fatty acid ratio.

**Table 10 animals-11-01259-t010:** Comparison of actual vs. predicted digestible energy (DE) of lipid sources fed to 13 kg and 50 kg pigs (adapted from [141]).

Lipid Source	Actual DE, Mcal/kg	Powles et al. [129] predicted DE ^1^, Mcal/kg	Kellner and Patience [141] Predicted DE ^2,3^, Mcal/kg
13 kg BW			
Animal–vegetable blend	8.81	8.40	8.34
Canola oil	8.59	8.82	8.56
Choice white grease source A	8.32	8.45	8.69
Choice white grease source B	8.67	8.46	8.79
Coconut oil	7.65	7.08	7.64
Corn oil source A	6.90	8.66	7.14
Corn oil source B	8.52	8.80	8.28
Fish oil	8.69	8.37	8.78
Flax oil	8.06	8.66	8.03
Palm oil	8.81	8.10	8.62
Poultry fat	8.67	8.57	8.38
Soybean oil source A	9.04	8.81	8.95
Soybean oil source B	8.99	8.81	8.97
Tallow	8.33	8.06	8.76
50 kg			
Animal–vegetable blend	7.51	8.40	7.69
Canola oil	9.53	8.82	9.52
Choice white grease source A	9.31	8.45	8.75
Choice white grease source B	8.72	8.46	8.77
Coconut oil	7.97	7.08	8.34
Corn oil source A	7.43	8.66	7.54
Corn oil source B	8.55	8.80	8.50
Fish oil	7.77	8.37	7.85
Flax oil	9.43	8.66	9.54
Palm oil	8.50	8.10	8.76
Poultry fat	8.14	8.57	7.93
Soybean oil source A	9.05	8.81	8.66
Soybean oil source B	8.18	8.81	8.71
Tallow	8.22	8.06	7.92

^1^ DE, kcal/kg = [36.898 − (0.005 × FFA, %) − 7.330 − 0.906 × unsaturated fatty acid:saturated fatty acid ratio]/0.004184. ^2^ For 13 kg pigs: DE, Mcal/kg = 9.363 − (0.097 × FFA, %) − (0.016 × omega-6:omega-3 fatty acid ratio) − (1.240 × arachidic acid, %) − (5.054 × insoluble impurities, %) + (0.014 × palmitic acid, %). ^3^ For 50 kg pigs: DE, Mcal/kg = 8.357 + (0.189 × unsaturated fatty acid: saturated fatty acid ratio) − (0.195 × FFA, %) − (6.768 × behenic acid, %) + (0.024 × PUFA, %).

**Table 11 animals-11-01259-t011:** Comparison of actual vs. predicted digestible energy (DE) content of fats and oils fed to 19 kg pigs (adapted from [144]).

Lipid Source	DE Actual, kcal/kg	DE Predicted ^1^, kcal/kg	DE Predicted ^2^, kcal/kg
Coconut oil	9380	7104	8518
Butter	8911	7496	8471
Tallow	8071	7828	8464
Palm oil	8304	7861	7595
Lard	8648	7968	8118
Fish oil	9464	8059	8524
Soybean oil	9979	8944	8769
Olive oil	9606	8947	8639
Flaxseed oil	8584	8873	7764
Canola oil	9474	9053	8589

^1^ Digestible energy predicted by [128]: DE, kcal/kg = [37.890 − (0.005 × FFA, g/kg of lipid) − 8.200e^(−0.515^
^× UFA:SFA)^]/0.004184. ^2^ Digestible energy predicted by [140]: DE, kcal/kg = [9.363 − (0.097 × FFA, %) − (0.016 × n-6:n-3 fatty acid ratio) − (1.240 × C20:0, %) − (5.054 × insoluble impurities, %) + (0.014 × C16:0, %)] × 1000.

**Table 12 animals-11-01259-t012:** Summary of one-, two-, and three-step closed in vitro filtration methods (adapted from [197]).

Method	Enzymes Used	References
1-step	Pepsin	[239]
	Trypsin	[240]
	Papain	[241]
	Pronase	[242]
2-step	Pepsin–Pancreatin	[243,244,245]
	Pepsin–Trypsin	[246]
	Pepsin–Pronase	[245]
	Pepsin–Jejunal fluid	[247]
3-step	Pepsin–Pancreatin–Cellulase	[248,249,250]
	Pepsin–Pancreatin–Viscozyme	[251,252]
	Pepsin–Pancreatin–Rumen fluid	[253]

**Table 14 animals-11-01259-t014:** Summary of functional ingredients, nutrients, bioactive compounds, and functions associated with improvements in swine health.

Ingredient	Bioactive Compounds	Functions	References
AAs	Glutamate, glutamine, glycine, proline, arginine	Signaling pathways regulating gene expression, intracellular turnover, nutrient metabolism, oxidative defense, and reducing intestinal damage	[171,333,334]
Animal plasma	Immunoglobulins	Improves immune response and gut-barrier function	[288,335,336]
Barley	Β-glucans, resistant starch, soluble and insoluble NSPs	Prebiotic to increase lactic acid production; improve gut health	[82,337,338,339,340]
Copper	Copper sulfate Tribasic copper chloride	Antibacterial alters gut microbiome	[341,342]
Essential fatty acids	Linoleic acid and linolenic acid	Improve reproductive performance in sows, affect inflammatory reactions and immune-response bacterial challenges and epithelial barrier function	[343,344,345]
Fermented liquid feed	Naturally occurring lactic acid bacteria and yeast	Production of lactic acid, acetic acid, and ethanol; reduces pH; prevents proliferation of pathogens	[338,346]
Fermented soybean meal	-	Decreased antinutritional factors, increased peptides, improves nutrient digestibility and gut microbiome	[347,348]
Functional fibers	Various types of NSPs	Alter gut microbiome, prebiotic, production of short-chain fatty acids, improve innate and adaptive immune responses, antioxidant and bactericidal properties	[82,83,89,90,94,349,350,351,352,353,354,355]
Lactose	Glucose and galactose	Improves nutrient digestibility, prebiotic, fermentation to lactic acid and volatile fatty acids in young pigs	[73,74,75,76]
Medium-chain fatty acids and monoglycerides	Caproic acid, caprylic acid, capric acid, and lauric acid; Glycerol monocaproate, glycerol monocaprylate, glycerol monocaprate, glycerol monolaurate	Antibacterial, antiviral, immune modulation activity, and improved gut health in pigs; feed-pathogen mitigation	[356,357,358,359,360,361,362]
Macroalgae–Seaweed	Functional fiber	Improve immune response and prebiotic; component of clay-based antimycotoxin agents	[363]
Microalgae	Omega-3 fatty acids	Improve immune response and reproduction; component of clay-based antimycotoxin agents	[364]
Oats	Β-glucans, resistant starch, soluble and insoluble NSPs	Prebiotic to increase lactic acid production; improve gut health	[82,337,338,339]
Soybean meal, soy-protein concentrate, and soy-protein isolate	Isoflavones	Anti-inflammatory, antioxidant, antiviral, decrease intestinal epithelial permeability, enhance growth and immune responses from PRRSV infection	[365,366]
Soybean meal, soy-protein concentrate, and soy-protein isolate	Saponins	Antioxidants, potential enhancers of passive immunity, vaccine adjuvants to increase immune response	[365]
Vitamins	A (carotenoids), C, D, E, K, niacin, pyridoxine, riboflavin	Antioxidants; gastrointestinal function and health	[367,368]
Zinc	Zinc oxide	Antibacterial, alters gut microbiome	[341,342]

**Table 15 animals-11-01259-t015:** Concentrations of total, insoluble, and soluble dietary fiber and phenolic acids in whole grain and bran fractions of common cereal grains (adapted from [372]).

Item	Barley		Corn		Oats		Rye		Wheat	
	Grain	Bran	Grain	Bran	Grain	Bran	Grain	Bran	Grain	Bran
Total dietary fiber, %	14.6–27.1	-	13.1–19.6	86.7	11.5–37.7	18.1–25.2	15.2–20.9	35.8	11.6–17.0	36.5–52.4
Insoluble dietary fiber, %	12.0–22.1	-	11.6–16.0	86.5	8.6–33.9	14.5–20.2	11.1–16.0	30.5	10.2–14.7	35.0–48.4
Soluble dietary fiber, %	2.6–5.0	-	1.5–3.6	0.2	2.9–3.8	3.6–5.0	3.7–4.5	5.3	1.4–2.3	1.5–4.0
Ferulic acid, mg/kg	168–723	2002–2017	380–1759	26,100–33,000	359	-	6–860	25–2780	4.5–1270	1942–5400
*p*-coumeric acid, mg/kg	4–374	2565–3367	31	3000–4000	-	-	41	100–190	0.2–37.2	100–457
Vanillic acid, mg/kg	29.2–33.4	82–117	4.6	-	17	-	3–22	10	0.6–35	100–164
Sinapic acid, mg/kg	-	-	57	-	55	-	2–120	53–100	1.3–63	300
Total phenolic content, mg gallic acid equivalent/kg	-	-	2194–3010	-	1223	1950	-	5840	350–1505	2800–5643

**Table 16 animals-11-01259-t016:** Summary of the potential role of vitamins in inhibiting enteric infections in pigs (adapted from [368]).

Vitamin	Mechanism	Impact
B-vitamins, E	Inhibition of inflammation via reduction of PGE2	Reduced inflammation provides a less favorable environment for ETEC
Carotenoids, C, D, E, K, niacin, pyridoxine, and riboflavin	Control oxidative stress	Minimize production of reactive oxygen species during an induced inflammatory response and prevention of enteric infection
A,B-complex, C, D, and E	Improved immune cell activity, response, and homeostasis	Formation of immune cells and signals and modulation of immune cell responses
A and D	Improved intestinal barrier function	Regulation of tight junction molecules and prevention of barrier damage
A and D	Regulation of innate and adaptive immunity and resolution of inflammation	Immune cell differentiation and cytokine suppression in response to injury and infection and resolution of inflammation
A and D	Production of antimicrobial peptides	Enhance innate immunity and composition of commensal microbiota
A, B_6_, B_12_, thiamin, riboflavin, C, D, E, and K	Affect microbiome composition	Regenerate commensal microbiota

## Data Availability

All data provided in this manuscript was adapted from other published sources and was appropriated cited in the tables, figures, and reference section.

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
