# Peer review of "Measures Matter—Determining the True Nutri-Physiological Value of Feed Ingredients for Swine"

_animals, 2021, doi:10.3390/ani11051259_

Round 1

Reviewer 1 Report

This review is very long and read more as a book chapter. Two different styles have been detected (both for semantic and font used) and it seems that this is a combination of the work of more authors that lack a final collation. Almost 400 citation is too much. Please revise thoroughly.

Author Response

We agree that this is a long review with numerous references. However, our goal was to make this a comprehensive review and combine several related but lesser-known topics that are seldom considered in a holistic manner. The font size and style of the manuscript was consistent when it was originally submitted to the journal, but apparently it was not checked by the journal editors when it was converted to a new format for the reviewers. We have standardized the format throughout the manuscript for consistency. You are correct that this is the work of 4 co-authors with different writing styles and semantics. We have made numerous revisions throughout the manuscript to improve and standardize semantics in this revision.

Reviewer 2 Report

In overall, the Manuscript has high relevance in terms of digestibility and availability of nutrients for pigs. However, I would suggest to the authors review carefully the whole Manuscript and to re-word some paragraphs that sound just copy and paste from other Manuscripts. 

Line 52: Define LCA before abbreviation

Line 123: Please check Font of whole Manuscript

Table 3 and Table 4: Tables are not fitted in the pages. Please review all tables. 

Author Response

In overall, the Manuscript has high relevance in terms of digestibility and availability of nutrients for pigs. However, I would suggest to the authors review carefully the whole Manuscript and to re-word some paragraphs that sound just copy and paste from other Manuscripts. 

Response: We have made substantial revisions throughout the entire manuscript as suggested.

Line 52: Define LCA before abbreviation

Response: Defined as suggested.

Line 123: Please check Font of whole Manuscript

Response: The font size and style of the manuscript was consistent when it was originally submitted but apparently was not checked by the journal editors when it was converted to a new format for the reviewers. We have standardized the format throughout the manuscript for consistency as suggested.

Table 3 and Table 4: Tables are not fitted in the pages. Please review all tables. 

Response: Tables 3, 4, and 5 were originally constructed in landscape format due to the spaces needs and was submitted in this format. However, when the journal editors converted it into a new format, these tables were placed in portrait format. We have converted these tables to landscape format again in the revised version to address this issue.

Reviewer 3 Report

The review manuscript is well organized and written with comprehensive and scientific views. It can be accepted for publication in Animals afer minor revision. In the section of 2.2 Energy systems, Effective metabolizable energy(EME) should be included and discussed, because it was raised in NRC (2012) and can be applied for diet formulation of pigs. There are some minors need to be revised. Abstract, Line 40,"One Health"is it correct?  Introduction, Line 51, (LCA) need to be added after "Life Cycle Assessments"; Line 152, "than" should be "that".

Author Response

The review manuscript is well organized and written with comprehensive and scientific views. It can be accepted for publication in Animals afer minor revision. In the section of 2.2 Energy systems, Effective metabolizable energy(EME) should be included and discussed, because it was raised in NRC (2012) and can be applied for diet formulation of pigs. There are some minors need to be revised. Abstract, Line 40,"One Health"is it correct?  Introduction, Line 51, (LCA) need to be added after "Life Cycle Assessments"; Line 152, "than" should be "that".

Response: Thank you for your comments.

A brief discussion was added on effective metabolizable energy as suggested.

We deleted “One Health” and replaced it with “…environmental sustainability in global pork production systems.”.

LCA was defined after “Life Cycle Assessments”.

Changed “than” to “that” as suggested.

Reviewer 4 Report

Title: Measures matter – Determining the true nutri-physiological value of feed ingredients for swine

Manuscript ID: animal-1146893

COMMENTS TO AUTHORS

GENERAL COMMENTS/QUESTIONS:

The authors reviewed different characteristics in feed ingredients fed to pigs. This manuscript is within the scope of Animals and reads well. Minor comments include:

This may be about editing. Please use the same size and font throughout the manuscript.

L 47-66: Not sure if they are related to the main context of this manuscript. This paragraph relates to more diseases of pigs and these are never mentioned later in this manuscript (e.g., ASFV or LCA).

L 94-97: Please revise this sentence.

L 51-52: Please define “LCA”.

L 54: Please change “has” to “have”.

L 118: Please change use “AA”, which has been defined previously.

L 177-181: Please add any reference(s) in which concentration of moisture was presented on an as-is basis and on a DM basis. It seems a bit odd that moisture content can be expressed on a DM basis.

L 190-192: Please revise this sentence. This sentence can be separated into 2 sentences.

L 201-202: The authors may want to add any comments on data in Table 1. For example, can be stated as “if the % moisture increased, the aw values increased, but with the same aw, the increment in the % moisture differed among the cereal grains”. The authors may want to relate these data to the reason why the aw should be used instead of % moisture.

Table 1: Please change “aw” to “Water activity, aw”.

Table 2: … “[54]”

Table 2: Was there no information on the particle sizes of corn? If so, please use the footnote to indicate this.

L 220: Please change to “[57]”.

L 260: The authors may want to use “to” instead of “for”.

L 420-441: Can the fact that ether extract concentrations do not vary much among diets or feed ingredients for pigs compared with other nutrients be one of the reasons?

L 542, 547: Please spell out “ADG”, “ADFI”.

L 697: Please add a reference “She et al. (2018); doi: 10.5713/ajas.17.0547”. This experiment tested the additivity of ATTD and STTD of P.

L 840: Please change to “… have improved…”.

Table 13: It seems that the title needs to be re-located.

References

I know organizing a tremendous number of references is time-consuming and tedious work, but the authors may want to recheck all the references. Sometimes they are not consistent.

#23: Please check this reference.

#56, 65, 117, 137, 142, 167, 173, 347: Please change to “Asian-Australas.”.

#135: Please use Italic for the journal name.

#208, 209, 303, 310, 345, 361, 362: Please check the page numbers. The page numbers should look like “txaa205”, “skaa097”, and so on.

#323: Please remove “1”.

Author Response

GENERAL COMMENTS/QUESTIONS:

The authors reviewed different characteristics in feed ingredients fed to pigs. This manuscript is within the scope of Animals and reads well. Minor comments include:

This may be about editing. Please use the same size and font throughout the manuscript.

Response: The font size and style of the manuscript was consistent when it was originally submitted but apparently was not checked by the journal editors when it was converted to a new format for the reviewers. We have standardized the format throughout the manuscript for consistency.

L 47-66: Not sure if they are related to the main context of this manuscript. This paragraph relates to more diseases of pigs and these are never mentioned later in this manuscript (e.g., ASFV or LCA).

Response: We have revised this paragraph to connect these concepts in a more concise manner.

L 94-97: Please revise this sentence.

Response: revised

L 51-52: Please define “LCA”.

Response: done

L 54: Please change “has” to “have”.

Response: done

L 118: Please change use “AA”, which has been defined previously.

Response: done

L 177-181: Please add any reference(s) in which concentration of moisture was presented on an as-is basis and on a DM basis. It seems a bit odd that moisture content can be expressed on a DM basis.

Response: a reference has been added

L 190-192: Please revise this sentence. This sentence can be separated into 2 sentences.

Response: sentence has been revised.

L 201-202: The authors may want to add any comments on data in Table 1. For example, can be stated as “if the % moisture increased, the aw values increased, but with the same aw, the increment in the % moisture differed among the cereal grains”. The authors may want to relate these data to the reason why the aw should be used instead of % moisture.

Response: Good suggestion. Additional text has been added.

Table 1: Please change “aw” to “Water activity, aw”.

Response: done

Table 2: … “[54]”

Response: revised

Table 2: Was there no information on the particle sizes of corn? If so, please use the footnote to indicate this.

Response: No particle size for corn was provided in this reference. A footnote to indicate this has been added as suggested.

L 220: Please change to “[57]”.

Response: done

L 260: The authors may want to use “to” instead of “for”.

Response: done

L 420-441: Can the fact that ether extract concentrations do not vary much among diets or feed ingredients for pigs compared with other nutrients be one of the reasons?

Response: Not really. Ether extract concentrations can vary significantly among some ingredients (e.g corn DDGS has 5 to 13% ether extract vs. corn grain with has ~3.5% ether extract). Furthermore, depending on the geographic location where swine diets are formulated, supplemental fats and oils are often added to North American diets resulting in greater ether extract content compared with Asian or European swine diets that may not contain any supplemental fat or oil.

L 542, 547: Please spell out “ADG”, “ADFI”.

Response: done

L 697: Please add a reference “She et al. (2018); doi: 10.5713/ajas.17.0547”. This experiment tested the additivity of ATTD and STTD of P.

Response: done

L 840: Please change to “… have improved…”.

Response: done

Table 13: It seems that the title needs to be re-located.

Response: yes, this is a formatting issue that has been corrected.

References

I know organizing a tremendous number of references is time-consuming and tedious work, but the authors may want to recheck all the references. Sometimes they are not consistent.

#23: Please check this reference.

Response: this reference has been checked and corrected.

#56, 65, 117, 137, 142, 167, 173, 347: Please change to “Asian-Australas.”.

Response: done

#135: Please use Italic for the journal name.

Response: corrected

#208, 209, 303, 310, 345, 361, 362: Please check the page numbers. The page numbers should look like “txaa205”, “skaa097”, and so on.

Response: These references have been checked and are correct as is. I think you may be referring to doi references which we did not include in this reference section.

#323: Please remove “1”.

Response: done

Round 2

Reviewer 1 Report

Please see comments